# IPDreamer: Appearance-Controllable 3D Object Generation with Complex Image Prompts

**Bohan Zeng**[1]*, **Shanglin Li**[4]*, **Yutang Feng**[4]*, **Ling Yang**[1]†, **Juan Zhang**[4]‡
**Hong Li**[1], **Jiaming Liu**[3], **Conghui He**[7], **Wentao Zhang**[1], **Jianzhuang Liu**[2]
**Baochang Zhang**[4]‡, **Shuicheng Yan**[5,6]
[1]Peking University  [2]Shenzhen Institute of Advanced Technology  [3]Tiamat AI
[4]Beihang University  [5]Skywork AI  [6]National University of Singapore
[7]Shanghai AI Laboratory

## Abstract

Recent advances in 3D generation have been remarkable, with methods such as DreamFusion leveraging large-scale text-to-image diffusion-based models to guide 3D object generation. These methods enable the synthesis of detailed and photorealistic textured objects. However, the appearance of 3D objects produced by such text-to-3D models is often unpredictable, and it is hard for single-image-to-3D methods to deal with images lacking a clear subject, complicating the generation of appearance-controllable 3D objects from complex images. To address these challenges, we present IPDreamer, a novel method that captures intricate appearance features from complex **I**mage **P**rompts and aligns the synthesized 3D object with these extracted features, enabling high-fidelity, appearance-controllable 3D object generation. Our experiments demonstrate that IPDreamer consistently generates high-quality 3D objects that align with both the textual and complex image prompts, highlighting its promising capability in appearance-controlled, complex 3D object generation. `https://github.com/zengbohan0217/IPDreamer`

## 1 Introduction

The rapid evolution of 3D technology has revolutionized the way we create and interact with virtual worlds. 3D technology is now essential in a wide range of fields, including architecture, gaming, mechanical manufacturing, and AR/VR. However, creating high-quality 3D content remains a challenging and time-consuming task, even for experts. To address this challenge, researchers have developed text-to-3D methodologies that automate the process of generating 3D assets from textual descriptions. Built on the 3D scene representation capabilities of Neural Radiance Fields (NeRFs) (Mildenhall et al., 2021; Müller et al., 2022) and the rich visual prior knowledge of pre-trained diffusion models (Rombach et al., 2022; Saharia et al., 2022a), recent research (Jain et al., 2022; Mohammad Khalid et al., 2022; Poole et al., 2022; Lin et al., 2023; Chen et al., 2023b; Wang et al., 2023; Shi et al., 2023b) has made significant progress, simplifying the text-to-3D pipeline and making it more accessible, which causes a significant shift in these fields.

Recent advances in diffusion models have significantly enhanced the capabilities of text-to-image generation. State-of-the-art (SOTA) systems, leveraging cutting-edge diffusion-based techniques (Nichol et al., 2021; Rombach et al., 2022; Brooks et al., 2023; Zhang & Agrawala, 2023; Hu et al., 2021), can now generate and modify images directly from textual descriptions with vastly improved quality. Inspired by the rapid development in text-to-image generation, recent works (Poole et al., 2022; Lin et al., 2023; Chen et al., 2023b; Wang et al., 2023) have extended these models to 3D by utilizing pretrained text-to-image diffusion models in conjunction with the Score Distillation Sampling (SDS) algorithm and its variants to optimize 3D representations. These methods are capable of

---

*These authors contributed equally.
†Project Leader.
‡Corresponding Author: bczhang@buaa.edu.cn, zhang_juan@buaa.edu.cn.

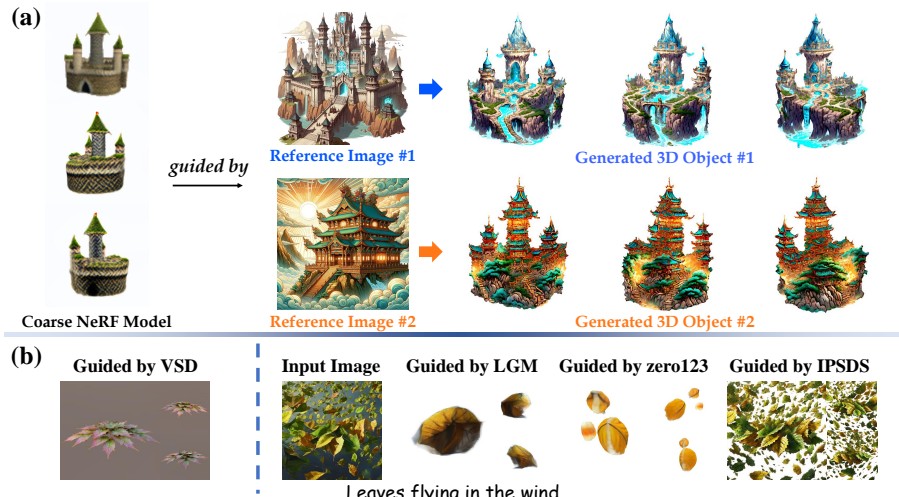

Figure 1: IPDreamer can generate controllable, high-quality 3D objects based on both textual and image prompts. (a) illustrates two high-quality 3D objects with rich details, initialized by the same NeRF model and guided by different complex reference image prompts. (b) demonstrates the 3D synthesis under challenging textual conditions, where our method outperforms existing text-to-3D method (Wang et al., 2023), image-to-3D methods LGM (Tang et al., 2024) and zero123++ (Shi et al., 2023a). Besides, the input image in (b) is generated by SD Rombach et al. (2022).

generating high-quality 3D objects and scenes. However, due to the lack of explicit appearance information in textual prompts, the appearance of the generated results remains largely uncontrollable, limiting the precision of the visual output.

Unlike the unpredictability in text-to-3D generation, single-image-to-3D generation allows for strict control over the appearance of the generated 3D results. However, existing single-image-to-3D methods (Liu et al., 2023b;c; Shi et al., 2023b) are limited to simple images with clear subjects, often struggling with complex images that feature rich content and intricate compositions. For example, Fig.1(a) contains complex images that lack a clear subject, making it difficult to segment a simple, singular image for single-image-to-3D generation. Furthermore, as demonstrated in Fig.1(b), when text prompts are ambiguous or lack a clear main subject—such as "Leaves flying in the wind"—neither current text-to-3D nor image-to-3D methods can achieve reasonable 3D synthesis.

To tackle these challenges, we introduce *IPDreamer*, a novel method for complex image-to-3D generation. Specifically, we extend SDS to *Image Prompt Score Distillation Sampling (IPSDS)*, which leverages detailed features extracted from complex image prompts and corresponding normal maps to guide the optimization of both 3D mesh texture and geometry. With IPSDS, IPDreamer enables high-quality 3D object generation with controllable appearances based on complex image inputs. To ensure stable 3D object generation across various challenging scenarios, we propose a mask-guided compositional alignment strategy for IPSDS, enabling 3D object generation from multiple complex image prompts. In particular, we leverage a Multimodal Large Language Model (MLLM) to localize the features from multiple image prompts onto the generated 3D objects. This allows IPDreamer to handle diverse situations, including cases where multiple highly divergent images are used to guide the synthesis of a single 3D object, and scenarios where the guiding complex image exhibits significant structural differences from the initial coarse 3D object. As shown in Fig.1(a), IPDreamer effectively transfers the appearances of reference images to NeRF models, generating high-quality 3D objects even in cases with ambiguities or unclear primary subjects. For the more challenging scenarios illustrated in Fig.1(b), IPDreamer successfully produces the desired 3D object where existing text-to-3D and single-image-to-3D methods fall short.

In summary, the main contributions of this paper are as follows:

- We present IPDreamer, a novel 3D object synthesis framework that allows users to consistently create controllable, high-quality 3D objects. Compared with previous methods,

IPDreamer excels in synthesizing high-quality 3D objects that closely align with complex image prompts.

- We introduce Image Prompt Score Distillation Sampling (IPSDS), which utilizes a substantial image prompt feature to guide 3D mesh optimization.

- We propose a Mask-guided Compositional Alignment strategy for IPSDS, enabling high-quality 3D objects synthesis based on multiple complex image prompts, in cases where initial NeRF models deviate significantly from the provided image prompts, or when multiple diverse image prompts are needed to guide the synthesis of a single 3D object.

- Comprehensive experiments show that IPDreamer achieves high-quality 3D generation and excellent rendering results, outperforming existing SOTA methods.

## 2 RELATED WORK

### 2.1 DIFFUSION MODELS

Diffusion models (DMs) were initially introduced as a generative model for gradually denoising images corrupted by Gaussian noise to generate samples (Sohl-Dickstein et al., 2015). Recent advancements in DMs (Ho et al., 2020; Song et al., 2020; Dhariwal & Nichol, 2021; Vahdat et al., 2021; Rombach et al., 2022; Peebles & Xie, 2022) have shown their exceptional performance in image synthesis. DMs have also achieved state-of-the-art results in various synthesis tasks, including text-to-image generation (Saharia et al., 2022a; Nichol et al., 2021; Ramesh et al., 2022; Podell et al., 2024; Yang et al., 2024), inpainting (Avrahami et al., 2022; Lugmayr et al., 2022; Ye et al., 2023), 3D object synthesis (Li et al., 2022b; Luo & Hu, 2021), video synthesis (Ho et al., 2022b;a), speech synthesis (Kong et al., 2020; Liu et al., 2021), super-resolution (Li et al., 2022a; Saharia et al., 2022b; Gao et al., 2023), face animation (Qi et al., 2023), text-to-motion generation (Tevet et al., 2022), and brain signal visualization (Takagi & Nishimoto, 2022; 2023). Some DMs (Kulikov et al., 2022; Wang et al., 2022) can produce diverse results by learning the internal patch distribution from a single image. (Mokady et al., 2023; Tumanyan et al., 2023; Wu et al., 2023; Geyer et al., 2023) enhance image/video editing with pre-trained DMs in a zero-shot or one-shot manner. These advancements highlight the versatility and potential of DMs across a wide range of syntheses.

### 2.2 CONTROLLABLE GENERATION AND EDITING

Controllable generation and editing of 2D images and 3D objects are core goals of generative tasks. With the emergence of large language models (LLMs) such as GPT-3 and Llama (Brown et al., 2020; Touvron et al., 2023a;b), instruction-based user-friendly generative control has gained much attention. InstructPix2Pix (Brooks et al., 2023) and MagicBrush (Zhang et al., 2023a) build datasets based on LLMs and large text-to-image models to achieve effective instruction control on 2D images. InstructNeRF2NeRF (Haque et al., 2023) combines this method with NeRF scene reconstruction (Mildenhall et al., 2021) to introduce instruction control into 3D generation. Meanwhile, a series of adapter methods such as ControlNet and IP-Adapter (Zhang & Agrawala, 2023; Hu et al., 2021; Mou et al., 2023; Zhao et al., 2023b; Ye et al., 2023; Huang et al., 2023) provide reliable approaches for fine-tuning large pre-trained DMs (e.g., Stable Diffusion (Rombach et al., 2022) and Imagen (Saharia et al., 2022a)) for conditional controllable generation (e.g., using sketch, canny, pose, etc. to control image structure). Among them, image prompt adaption methods (Ye et al., 2023; Zhang et al., 2023b) introduce a decoupled cross-attention mechanism to achieve effective appearance generation control using image prompts.

### 2.3 3D GENERATION

In recent years, 3D generative modeling has attracted a large number of researchers. Inspired by the recent neural volume rendering, many 3D-aware image rendering methods (Chan et al., 2022; 2021; Gu et al., 2021; Hao et al., 2021; Niemeyer & Geiger, 2021) are proposed to generate high-quality rendered 2D images for 3D visualization. Meanwhile, with the development of text-to-image synthesis, researchers have shown a growing interest in text-to-3D generation. Early methods such as DreamField (Jain et al., 2022) and CLIPmesh (Mohammad Khalid et al., 2022) achieve text-to-3D generation by utilizing a pretrained image-text aligned model CLIP (Radford et al., 2021). They

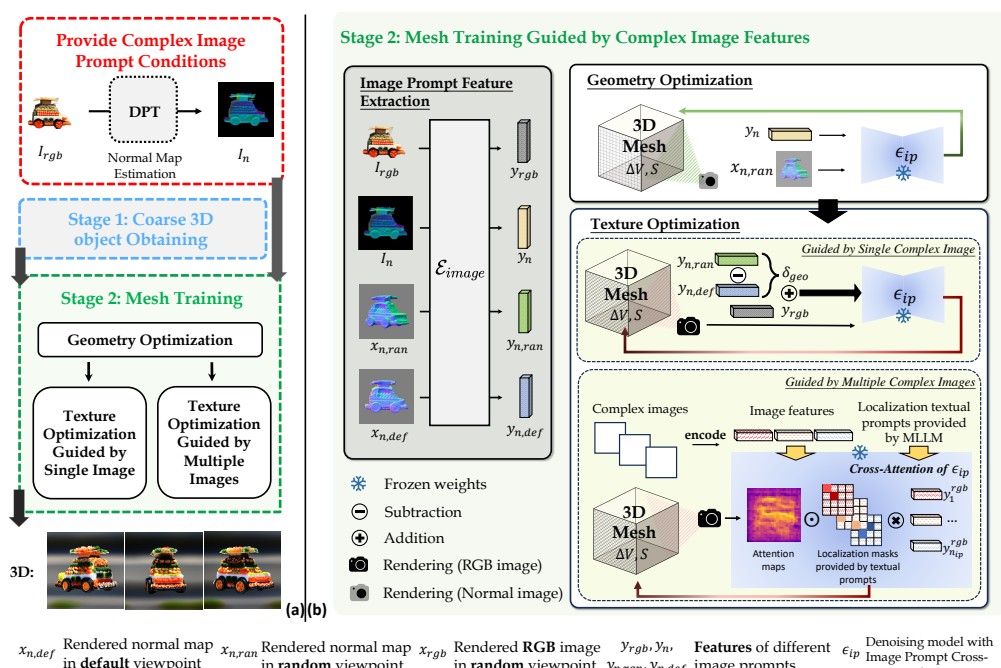

Figure 2: IPDreamer is designed to generate high-quality, appearance-controllable 3D meshes that align with single/multiple complex image prompts.

optimize the underlying 3D representations (NeRFs and meshes) to ensure that all 2D renderings have high text-image alignment scores. Recently, (Poole et al., 2022; Lin et al., 2023; Chen et al., 2023b; Wang et al., 2023; Chen et al., 2023a) have achieved high-quality 3D synthesis (NeRFs and meshes) by leveraging a robust pretrained text-to-image DM as a strong prior to guiding the training of the 3D model. Other works (Shi et al., 2023b; Zhao et al., 2023a; Liu et al., 2023b) introduce multi-view DMs to enhance 3D consistency and provide strong structured semantic priors for 3D synthesis. IT3D (Chen et al., 2023c) combines SDS and GAN to refine the 3D model and obtain high-quality 3D synthesis. (Tang et al., 2023a; Liang et al., 2023) combine 3D Gaussians (Kerbl et al., 2023) with SDS-based optimization to improve 3D synthesis and reduce generation time. Additionally, (Melas-Kyriazi et al., 2023; Tang et al., 2023b; Liu et al., 2023a; Qian et al., 2023) are capable to generate 3D representations based on single images, and (Liu et al., 2023b;c; Shi et al., 2023b; Yang et al., 2023) achieve 2D images in multiple viewpoints, which enable consistant 3D object generation. In this work, we introduce IPDreamer, a method that leverages complex image prompts to provide comprehensive appearance information, effectively guiding the synthesis of high-quality 3D objects.

## 3 METHOD

In this section, we present the details of IPDreamer. We start with a brief definition of 3D mesh, followed by the problem statement for text-to-3D and single-image-to-3D generation, and a review of SDS preliminaries. Next, we present the design and analysis of our proposed IPSDS and Mask-guided Compositional Alignment. Note that 3D meshes are the most common form of 3D representation in industry, so our approach focuses on optimizing 3D meshes.

### 3.1 PRELIMINARIES AND MOTIVATION

**3D Mesh.** The 3D mesh can be represented as a deformable tetrahedral grid $(V, T)$, where each vertex $v_i \in V$ has a signed distance field value $s_i \in S$ and a deformation $\Delta v_i \in \Delta V$ from its canonical position. During optimization, the surface mesh is rendered into high-resolution images using a differentiable rasterizer (Munkberg et al., 2022).

**Score Distillation Sampling.** Given a textual prompt $y$ or an image $I$, text-to-3D/single-image-to-3D generation aims to synthesize novel views and optimize the parameters of a 3D object/scene

corresponding to the given $y$ or $I$. DreamFusion (Poole et al., 2022) utilizes a pretrained text-to-image DM $\epsilon_{pretrain}$ to optimize an MLP parameterized as $\theta$ representing a 3D volume, where a differentiable generator $g$ renders $\theta$ to create 2D images $x = g(\theta, c)$ given a sampled camera pose $c$, based on the gradient of the Score Distillation Sampling (SDS) loss:

$$\nabla_\theta \mathcal{L}_{\text{SDS}}(\theta) = \mathbb{E}_{t,\epsilon} \left[ w(t) \left( \epsilon_{pretrain}(x_t; y, t) - \epsilon \right) \frac{\partial x}{\partial \theta} \right], \tag{1}$$

where $w(t)$ is a weighting function, $\epsilon_{pretrain}(x_t; y, t)$ predicts the noise $\epsilon \sim \mathcal{N}(0, \text{I})$, given the noisy image $x_t$, text prompt features $y$ and timestep $t$.

**Motivation.** Although (Lin et al., 2023; Chen et al., 2023b; Wang et al., 2023) show excellent text-to-3D generation, the appearances of their 3D synthesis results are uncontrollable. A feasible solution to realize controllable 3D object generation is to use a 2D image as a prior. However, existing single-image-to-3D methods find obtaining high-quality 3D object synthesis from complex images difficult. To overcome these obstacles, we extract feature details from complex image prompts to provide comprehensive appearance information for 3D object synthesis, and we propose the IPSDS and a Mask-guided Compositional Alignment strategy to enable stable, high-fidelity 3D object generation.

## 3.2 Image Prompt Score Distillation Sampling (IPSDS)

In this section, we leverage the image features extracted by the image encoder $\mathcal{E}_{image}$ from (Ye et al., 2023) to optimize the geometry and texture of 3D objects. The 3D mesh is initialized using either a user-provided 3D mesh, or a coarse NeRF model generated by existing text-to-3D methods or IPSDS. As shown in Fig. 2(b), we first explain how IPSDS optimizes $\Delta V$, $S$, and $\theta$. Subsequently, we analyze how IPSDS effectively utilizes complex image prompts $I_{rgb}$ and corresponding normal image prompts $I_n$ to guide the synthesis of high-quality 3D objects.

**Optimizing 3D Mesh with IPSDS.** Existing methods directly use a text-conditioned DM to guide geometry optimization. However, it can be challenging because the DM's pre-training dataset lacks normal map images. To address this, we adopt an additional normal image prompt feature $y_n = \mathcal{E}_{image}(I_n)$ to provide richer and more robust geometric information with normal map optimization, instead of solely using the textual prompt $y$ (Chen et al., 2023b; Wang et al., 2023).

The geometry optimization process computes the gradients of the IPSDS geometry loss as:

$$\nabla_{\Delta V} \mathcal{L}_{\text{IPSDS}-Geo}(\Delta V, S) = \mathbb{E}_{t,\epsilon}[w(t) \left( \epsilon_{ip}(z_{n,t}; y_n, y, t) - \epsilon \right) \frac{\partial z_n}{\partial \Delta V}], \tag{2}$$

$$\nabla_S \mathcal{L}_{\text{IPSDS}-Geo}(\Delta V, S) = \mathbb{E}_{t,\epsilon}[w(t) \left( \epsilon_{ip}(z_{n,t}; y_n, y, t) - \epsilon \right) \frac{\partial z_n}{\partial S}], \tag{3}$$

where $z_{n,t}$ denotes the noisy latent of rendered normal map in random view $x_{n,ran}$ at timestep $t$.

After optimizing the estimated normal map, the geometry of the 3D mesh becomes more reasonable. Then we further optimize the texture through IPSDS. We first extract the image prompt features $y_{rgb} = \mathcal{E}_{image}(I_{rgb})$, as a basic guidance for the texture optimization. Then we devise a geometry prompt difference $\delta_{geo}$ for $y_{rgb}$ to compensate for the morphological disparity between $x_{rgb}$ and $I_{rgb}$. Let $x_{n,def}$, and $x_{n,ran}$ be the rendered normal map of the 3D mesh from the default viewpoint and a randomly sampled viewpoint, respectively. We extract their image prompt featuress, $y_{n,def} = \mathcal{E}_{image}(x_{n,def})$ and $y_{n,ran} = \mathcal{E}_{image}(x_{n,ran})$. The difference between $y_{n,ran}$ and $y_{n,def}$ is called the geometry prompt difference $\delta_{geo}$:

$$\delta_{geo} = y_{n,ran} - y_{n,def}. \tag{4}$$

The texture optimization process computes the gradients of the IPSDS texture loss as:

$$\nabla_\theta \mathcal{L}_{\text{IPSDS}-Tex}(\theta, \Delta V, S) = \mathbb{E}_{t,\epsilon}[w(t) \left( \epsilon_{ip}(z_{rgb,t}; y_{rgb} + \delta_{geo}, y, t) - \epsilon \right) \frac{\partial z_{rgb}}{\partial \theta}], \tag{5}$$

$$\nabla_{\Delta V} \mathcal{L}_{\text{IPSDS}-Tex}(\theta, \Delta V, S) = \mathbb{E}_{t,\epsilon}[w(t) \left( \epsilon_{ip}(z_{rgb,t}; y_{rgb} + \delta_{geo}, y, t) - \epsilon \right) \frac{\partial z_{rgb}}{\partial \Delta V}], \tag{6}$$

$$\nabla_S \mathcal{L}_{\text{IPSDS}-Tex}(\theta, \Delta V, S) = \mathbb{E}_{t,\epsilon}[w(t) \left( \epsilon_{ip}(z_{rgb,t}; y_{rgb} + \delta_{geo}, y, t) - \epsilon \right) \frac{\partial z_{rgb}}{\partial S}], \tag{7}$$

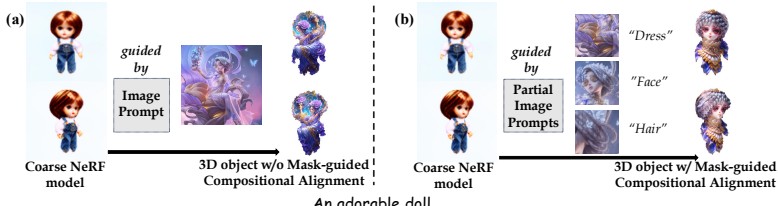

Figure 3: Illustration of the effectiveness of Mask-guided Compositional Alignment.

where $z_{rgb,t}$ denotes the noisy latent of $x_{rgb}$ (random viewpoint) in timestep $t$. The geometry prompt difference $\delta_{geo}$ can effectively represent the Morphological distance between $x_{n,ran}$ and $x_{n,def}$ in the image prompt feature space. Thus it is used to compensate $y_{rgb}$ (default viewpoint) such that $y_{rgb} + \delta_{geo}$ represents the RGB image $x_{rgb}$.

**Incorporating Image Prompt into 3D Generation.** Here we explain how our method can effectively use a complex, high-quality image to guide 3D object synthesis, by introducing cross-attention for the image prompt. Given the query features $Z$ which are derived from the latent representations of the 2D rendering results of the 3D object from various viewpoints, and the image prompt embedding $y_{rgb}$, the cross-attention for the image prompt is formulated as follows:

$$Z' = \text{Softmax}(\frac{\mathbf{Q}\mathbf{K}^\top}{\sqrt{d}})\mathbf{V}, \tag{8}$$

where $\mathbf{Q} = Z\mathbf{W}_q$, $\mathbf{K} = y_{rgb}\mathbf{W}_k^{ip}$, $\mathbf{V} = y_{rgb}\mathbf{W}_v^{ip}$ represent the queries, keys, and values within the cross-attention module, respectively, $Z'$ denotes the output features of the module, and the $\mathbf{W}_q$, $\mathbf{W}_k^{ip}$, and $\mathbf{W}_v^{ip}$ are the projection matrices used for linear transformations. The reasons why IPSDS can utilize complex images to guide the generation of 3D objects while existing single-image-to-3D methods cannot are two-fold: First, the encoder of the image prompt adaption method effectively extracts the image features $y_{rgb}$ from the reference high-resolution image prompt. Secondly, as the attention map can accurately align the features $y_{rgb}$ with specific positions of rendered images (Hertz et al., 2022) from the 3D object, the features from the original complex image are precisely positioned on the most relevant parts of the 3D object.

### 3.3 MASK-GUIDED COMPOSITIONAL ALIGNMENT FOR MULTIPLE IMAGE PROMPTS

**Motivation of Mask-guided Compositional Alignment.** When the rendered 2D images from the NeRF model significantly differ from the image prompt or when the input includes multiple diverse image prompts, relying solely on cross-attention for the image prompt fails to effectively align the features $y_{rgb}$ with specific positions on the 3D object. As illustrated in Fig. 3(a), it is evident that IPDreamer encounters difficulties with this challenging sample. To address this issue, we have designed a Mask-guided Compositional Alignment strategy. Specifically, we collect multiple images $I_i^{rgb}$ from the input complex image prompts. Then, we employ a large multimodel model (GPT-4v) to provide localization words $y_i^{txt}$ for corresponding $I_i^{rgb}$. We adopt the cross attention in LDM (Rombach et al., 2022) to obtain localization masks:

$$m_i = \text{BI}(\text{Softmax}(\frac{\mathbf{Q}\mathbf{K}_{txt,i}^\top}{\sqrt{d}})), \ \mathbf{Q} = Z\mathbf{W}_q, \ \mathbf{K}_{txt,i} = y_i^{txt}\mathbf{W}_k^{txt}, \ i = 1, 2, ..., n_{ip}, \tag{9}$$

where $Z$ represents the same query features as in Equation 8, BI denotes a binarization operator and $n_{ip}$ is the number of the input multiple images. Subsequently, the mask $m_i$, obtained from the textual description $y_i^{txt}$, is used to adjust the computation of the cross attention corresponding to the feature $y_i^{rgb}$ of the image prompt $I_i^{rgb}$:

$$Z' = \frac{1}{n_{ip}} \sum_{i=1}^{n_{ip}} m_i \, \text{Softmax}(\frac{\mathbf{Q}\mathbf{K}_{ip,i}^\top}{\sqrt{d}})\mathbf{V}_{ip,i}, \tag{10}$$

where $\mathbf{Q} = Z\mathbf{W}_q$, $\mathbf{K}_{ip,i} = y_i^{rgb}\mathbf{W}_k^{ip}$, $\mathbf{V}_{ip,i} = y_i^{rgb}\mathbf{W}_v^{ip}$. With the help of our strategy, we modify the cross-attention calculation during the IPSDS supervision process, enabling the features of multiple images to be localized onto the 3D object, as shown in Fig. 3(b). Next, we provide more details of the IPSDS training process with the Mask-guided Compositional Alignment.

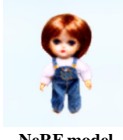 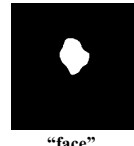 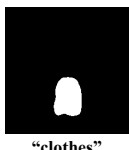 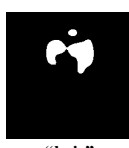

NeRF model      "face"      "clothes"      "hair"

Figure 4: Visualization of localization masks.

**Training Process of IPSDS with Mask-guided Compositional Alignment.** We briefly describe how IPSDS is designed with the Mask-guided Compositional Alignment. First, we utilize GPT-4v to generate localization textual prompts, $y_1^{txt}, ..., y_{n_{ip}}^{txt}$, to map features of the complex images onto 3D objects. Specifically, we input the multiple complex reference images and rendered images of a coarse NeRF model into GPT-4v, which analyzes and identifies the regions that need to be segmented from the input complex images and generates the corresponding localization textual prompts. Based on the analysis, we employ SAM (Kirillov et al., 2023) to segment multiple partial images, $I_1^{rgb}, ..., I_{n_{ip}}^{rgb}$, from the complex images. Additionally, both the localization textual prompts, $y_1^{txt}, ..., y_{n_{ip}}^{txt}$, and the segmented partial images, $I_1^{rgb}, ..., I_{n_{ip}}^{rgb}$, can be adjusted by users.

Given potential semantic differences between $I_{rgb}$ and the coarse NeRF model (e.g., "magnificent magic castle" vs. "adorable cottage") and the possibility that the multiple partial images $I_1^{rgb}, ..., I_{n_{ip}}^{rgb}$ may lack detail or resolution, it is crucial to enhance them before initiating texture optimization. We adopt a super-resolution model (Zhang & Agrawala, 2023) [1] in conjunction with $I_1^{rgb}, ..., I_{n_{ip}}^{rgb}$ and $y_1^{txt}, ..., y_{n_{ip}}^{txt}$ to generate new $I_1^{rgb}, ..., I_{n_{ip}}^{rgb}$. This preprocessing step improves the quality of both the guided images and the resulting 3D object.

Subsequently, we extract image prompt features $y_1^{rgb}, ..., y_{n_{ip}}^{rgb}$ from corresponding partial image prompts $I_1^{rgb}, ..., I_{n_{ip}}^{rgb}$. Then, we modify the cross-attention calculation based on Equation 9 and Equation 10 during the IPSDS supervision process, to localize the image features onto the 3D object according to $y_1^{txt}, ..., y_{n_{ip}}^{txt}$. Fig. 4 shows an example of the effect of localization masks calculated in the process of Mask-guided Compositonal Alignment. The IPSDS supervision in this part can be written as:

$$\nabla_\theta \mathcal{L}_{\text{IPSDS}-Tex}(\theta, \Delta V, S) = \mathbb{E}_{t,\epsilon}[w(t)\,(\epsilon_{ip}(z_{rgb,t}; y_1^{rgb}, ..., y_{n_{ip}}^{rgb}, y_1^{txt}, ..., y_{n_{ip}}^{txt}, t) - \epsilon)\frac{\partial z_{rgb}}{\partial \theta}], \qquad (11)$$

$$\nabla_{\Delta V} \mathcal{L}_{\text{IPSDS}-Tex}(\theta, \Delta V, S) = \mathbb{E}_{t,\epsilon}[w(t)\,(\epsilon_{ip}(z_{rgb,t}; y_1^{rgb}, ..., y_{n_{ip}}^{rgb}, y_1^{txt}, ..., y_{n_{ip}}^{txt}, t) - \epsilon)\frac{\partial z_{rgb}}{\partial \Delta V}], \quad (12)$$

$$\nabla_S \mathcal{L}_{\text{IPSDS}-Tex}(\theta, \Delta V, S) = \mathbb{E}_{t,\epsilon}[w(t)\,(\epsilon_{ip}(z_{rgb,t}; y_1^{rgb}, ..., y_{n_{ip}}^{rgb}, y_1^{txt}, ..., y_{n_{ip}}^{txt}, t) - \epsilon)\frac{\partial z_{rgb}}{\partial S}]. \qquad (13)$$

After initially localizing the partial image prompts onto the 3D object, it is then necessary to further optimize the texture of the 3D object globally. We input all features of the partial images and the provided complex images into the IPSDS loss to optimize the 3D object simultaneously:

$$f_{global} = \text{concat}(y_1^{rgb}, ..., y_{n_{ip}}^{rgb}, y_{rgb} + \delta_{geo}), \qquad (14)$$

$$\nabla_\theta \mathcal{L}_{\text{IPSD}-Tex}(\theta, \Delta V, S) = \mathbb{E}_{t,\epsilon}[w(t)(\epsilon_{ip}(z_{rgb,t}; f_{global}, t) - \epsilon)\frac{\partial z_{rgb}}{\partial \theta}], \qquad (15)$$

$$\nabla_{\Delta V} \mathcal{L}_{\text{IPSD}-Tex}(\theta, \Delta V, S) = \mathbb{E}_{t,\epsilon}[w(t)(\epsilon_{ip}(z_{rgb,t}; f_{global}, t) - \epsilon)\frac{\partial z_{rgb}}{\partial \Delta V}], \qquad (16)$$

$$\nabla_S \mathcal{L}_{\text{IPSD}-Tex}(\theta, \Delta V, S) = \mathbb{E}_{t,\epsilon}[w(t)(\epsilon_{ip}(z_{rgb,t}; f_{global}, t) - \epsilon)\frac{\partial z_{rgb}}{\partial S}]. \qquad (17)$$

## 4 EXPERIMENTS

### 4.1 3D GENERATION WITH SINGLE COMPLEX IMAGE

As depicted in Fig. 5, we show generated 3D objects that use diverse image prompts to guide synthesis. This demonstrates IPDreamer's ability to produce high-quality 3D objects that align with

---

[1] https://huggingface.co/lllyasviel/control_v11f1e_sd15_tile

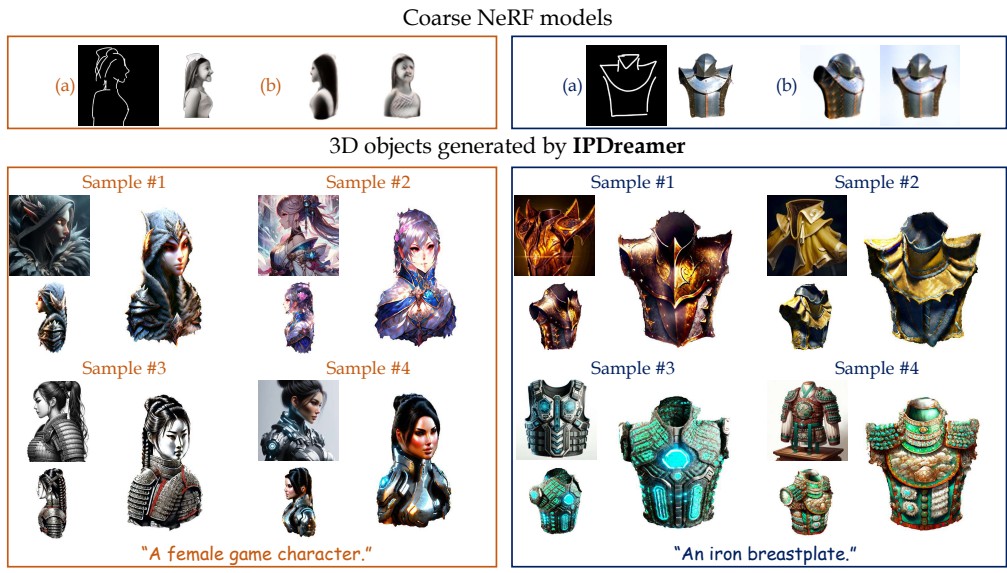

Figure 5: generated 3D objects with different image prompts. (a) Image prompts used for Coarse NeRF model generation. (b) Rendering of Coarse NeRF models. We show four samples for each textual prompt. In each sample, the top left is a selected complex image prompt, and the bottom left and the right illustrate the high-quality 3D object optimized by IPDreamer based on the coarse NeRF model.

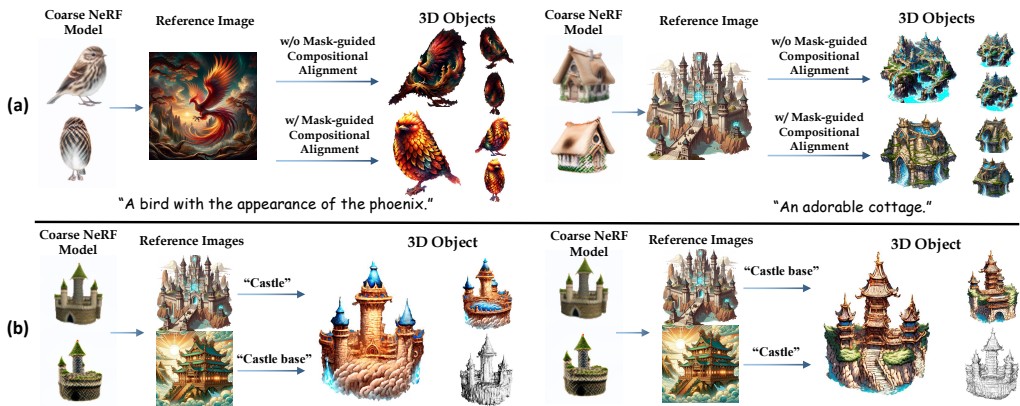

Figure 6: Effectiveness of Mask-guided Compositional Alignment.

the styles of the provided images. Remarkably, IPDreamer can appropriately transfer the appearance of the image prompts to the synthesized 3D objects, regardless of the structure difference between the image prompts and the coarse NeRF models. To our knowledge, this high-quality appearance transfer task is not achievable by existing text-to-3D or single-image-to-3D methods. In Sample 2 for the textual prompt "An iron breastplate", although both the textual and image prompt features are provided for 3D object synthesis, the generated result resembles a leather breastplate more closely, which aligns with the image prompt rather than the "iron" mentioned in the textual prompt. This illustrates that the image prompt exerts a stronger influence on the synthesis of the 3D object than the textual prompt. Consequently, such a powerful ability to edit 3D object textures greatly facilitates applications in the gaming and video industries.

## 4.2 3D GENERATION WITH MULTIPLE COMPLEX IMAGES

To demonstrate the stability of our IPDreamer in generating 3D models guided by multiple complex images or when the initial coarse 3D object significantly differs from these guiding images, we produced more 3D objects under these conditions. As shown in Fig. 6(a), when provided im-

Table 1: Quantitive comparison of text-to-3D generation.

| Method | FID ↓ | CLIP-Score ↑ |
|---|---|---|
| DreamFusion | 320.16 | 0.2413 |
| Magic3D | 320.56 | 0.2582 |
| Fantasia3D | 294.79 | 0.2557 |
| ProlificDreamer | 277.35 | 0.2603 |
| LRM | 304.81 | 0.2522 |
| LGM | 296.62 | 0.2605 |
| Zero123++ | 269.58 | 0.2561 |
| SV3D | 266.21 | 0.2681 |
| IPDreamer(Ours) | **253.32** | **0.2716** |

Table 2: Percentage of the preference in the user study of text-to-3D generation.

| Method | Prefer baseline | Prefer ours |
|---|---|---|
| DreamFusion | 6.45 | **93.55** |
| Magic3D | 10.89 | **89.11** |
| Fantasia3D | 25.82 | **72.18** |
| ProlificDreamer | 41.65 | **58.35** |
| LRM | 29.73 | **70.27** |
| LGM | 34.52 | **65.48** |
| zero123++ | 33.75 | **66.25** |
| SV3D | 43.75 | **56.25** |

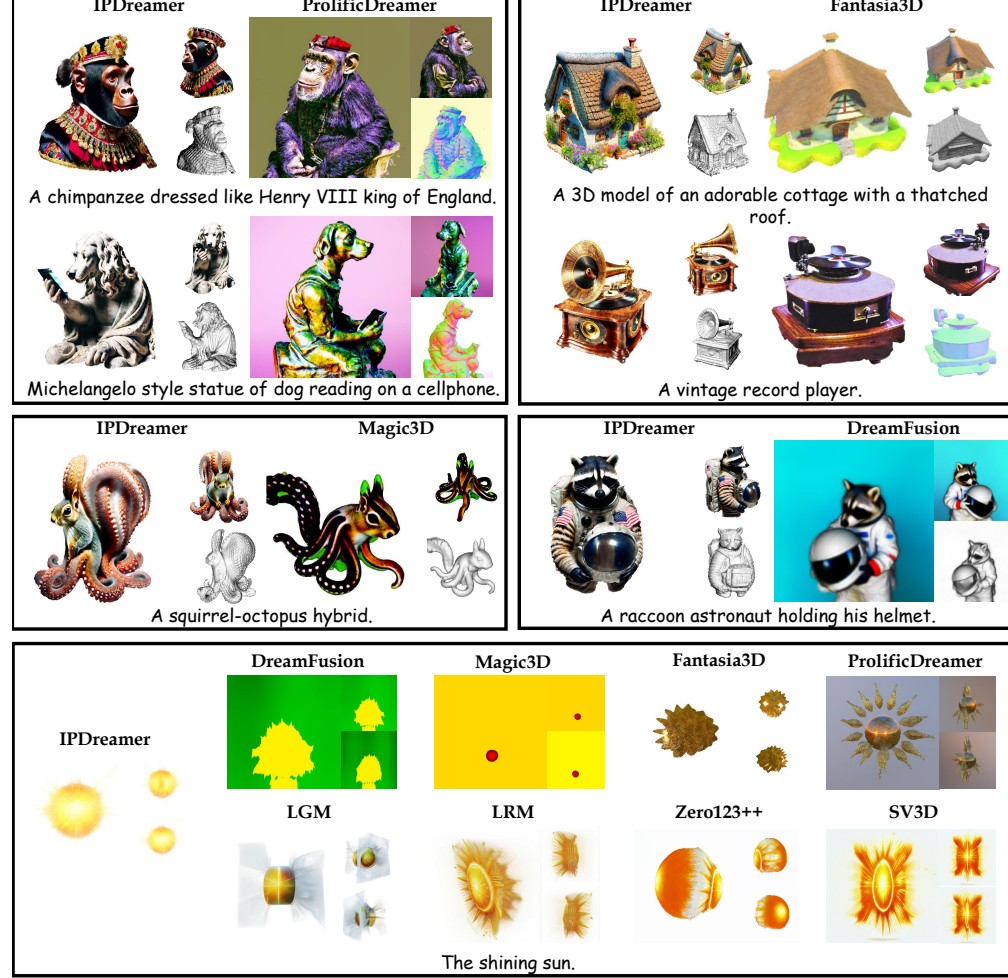

Figure 7: Qualitative Comparison of Text-to-3D Generation. It is worth noting that for the "Shining Sun" sample, our IPDreamer can generate a luminous 3D sphere with natural light rays emitting, which is difficult for other baseline methods to achieve.

age prompts vastly differ from the coarse NeRF models, the 3D objects guided by IPSDS with Mask-guided Compositional Alignment retain the semantic essence of the original coarse NeRF models and achieve the intended outcomes. And in Fig. 6(b), we provide two samples with the same coarse NeRF model and multiple diverse complex image prompts but different mask-guided textual prompts; the generated results of these two samples are quite different and well follow the

w/o $\mathcal{L}_{\text{IPSDS}-Geo}$    **Sample 1**    w/ $\mathcal{L}_{\text{IPSDS}-Geo}$      **Sample 2**    w/ $\mathcal{L}_{\text{IPSDS}-Geo}$

Figure 8: Visualization of the initial normal map of the 3D object at the beginning of geometry optimization, along with the image prompt and the refined normal map after geometry optimization for each sample.

Table 3: Ablation study of $\delta_{geo}$.

| Methods | CLIP score ↑ |
|---|---|
| w/o $\delta_{geo}$ | 0.8228 |
| w/ $\delta_{geo}$ | **0.8389** |

textual requirements, showing that IPDreamer effectively enhances the diversity of the generated 3D objects, offering new perspectives for the advancement in the 3D research domain.

### 4.3 COMPARISON ON TEXT-TO-3D

To validate the quality of the results generated by our method, we conducted a comparative analysis with text-to-3D methods (Poole et al., 2022; Lin et al., 2023; Chen et al., 2023b; Wang et al., 2023) and single-image-to-3D methods (Hong et al., 2024; Tang et al., 2024; Shi et al., 2023a; Voleti et al., 2025) in the text-to-3D generation task. As illustrated in Fig. 7, IPDreamer surpasses these baseline methods by producing highly controllable and realistic 3D objects that align closely with the provided textual prompts. Additionally, we compare IPDreamer with all baseline methods under example "The shining sun", where existing text-to-3D and single-image-to-3D methods fail to generate clear and coherent subjects. In contrast, our method successfully generates results that meet the requirements, further demonstrating its effectiveness.

For a quantitative evaluation, we randomly select 30 textual prompts and compare the performance of IPDreamer against state-of-the-art (SOTA) methods, as shown in Table 1. IPDreamer achieves superior performance, evidenced by a lower FID score, indicating higher quality 3D object generation, and a higher CLIP score, reflecting better alignment with the input textual prompts. To provide a more comprehensive assessment of the generated results, we also conduct a user study, the results are demonstrated in Table. 2. The details of the CLIP score, FID, and user study are introduced in Appendix A.2. Besides, we provide more generated 3D results in Appendix A.1.3.

### 4.4 ABLATION STUDY

We conduct an ablation study to evaluate the impact of $\mathcal{L}_{\text{IPSDS}-Geo}$ and $\delta_{geo}$ on optimizing 3D objects. Their effectiveness is illustrated in Fig. 8 and Table 3. In Fig. 8, we showcase the optimized normal maps of two samples. After geometry optimization, Sample 1 and Sample 2 learn the high-frequency details from their corresponding image prompts. The difference in the optimized normal maps between Sample 1 and Sample 2 is readily discernible in Fig. 8, illustrating the efficacy of $\mathcal{L}_{\text{IPSDS}-Geo}$ in learning geometry representations from image prompts. In Table 3, we compare the CLIP score of 3D objects optimized with and without $\delta_{geo}$. We conduct the quantitive comparison using the samples mentioned in Section 4.3 and employ CLIP score to compare the alignment of rendered images of 3D objects generated with and without $\delta_{geo}$ in different viewpoints with the reference image prompt. The experimental results show that with $\delta_{geo}$, the rendered images of the 3D object in different viewpoints are more consistent with the reference image prompt.

### 5 CONCLUSION

In this work, we propose IPDreamer, a novel framework that enables the generation of high-quality, appearance-controllable 3D objects from complex image prompts. By introducing Image Prompt Score Distillation Sampling (IPSDS), our method effectively captures rich and intricate appearance features from complex images to guide the optimization of both texture and geometry in 3D mesh generation. Our approach supports multiple complex images in various contexts to guide 3D object generation, enabling the stable production of high-quality 3D results. IPDreamer addresses the limitations of existing text-to-3D and single-image-to-3D methods by producing 3D objects that are consistent with textual descriptions and the detailed appearances of complex image prompts. Comprehensive experiments demonstrate that IPDreamer outperforms state-of-the-art methods, highlighting its promising capability in advancing appearance-controllable complex 3D object generation.

## 6 ACKNOWLEDGEMENTS

The work was supported by the National Key Research and Development Program of China (Grant No. 2023YFC3306401). This research was also supported by Zhejiang Provincial Natural Science Foundation of China under Grant No. LD24F020007, Beijing Natural Science Foundation L223024, L244043 and Z241100001324017, "One Thousand Plan" projects in Jiangxi Province Jxsq2023102268, National Natural Science Foundation of China under Grant No. NSFC12201024, and in part by NUS Start-up Grant A-0010106-00-00.

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

## A APPENDIX

In Appendix A.1, we present additional synthesized 3D objects generated by IPDreamer. Detailed implementation information is provided in Appendix A.2. Furthermore, Appendix A.5 analyzes the social impact of IPDreamer.

### A.1 MORE EXAMPLES OF 3D OBJECTS GENERATED BY IPDREAMER

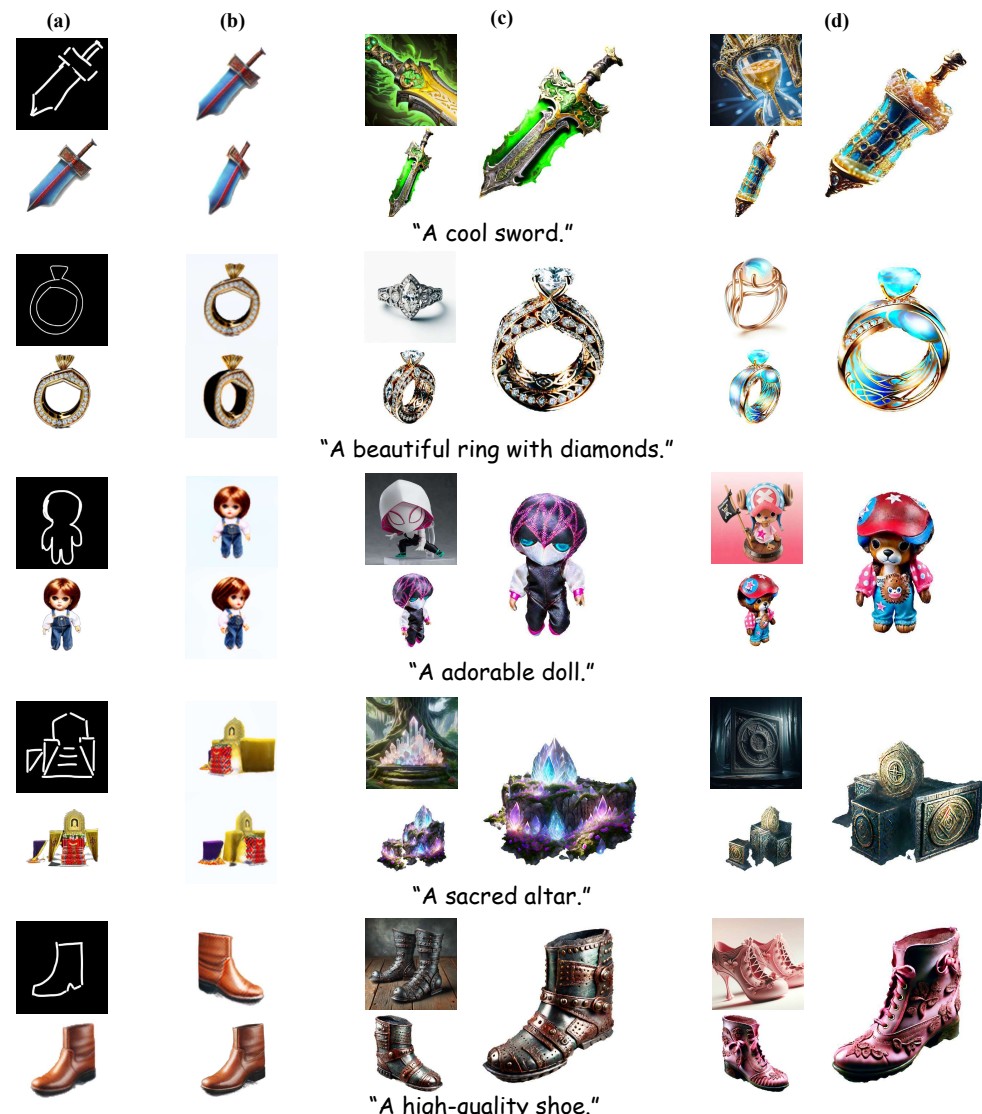

Figure 9: generated 3D objects with different image prompts. (a) Scribble object outlines and corresponding image prompts for Coarse 3D object generation. (b) Renderings of coarse NeRF models. (c) (d) Two samples demonstrated for each textual prompt. In each sample, the top left is a reference complex image prompt, and the bottom left and the right illustrate the 3D object optimized by IPDreamer based on the coarse NeRF model.

### A.1.1 MORE EXAMPLES OF 3D OBJECTS GUIDED BY IPSDS

To further demonstrate IPDreamer's remarkable ability to manipulate appearance, we conduct more 3D object synthesis experiments. These experiments use diverse textual prompts, each accompanied

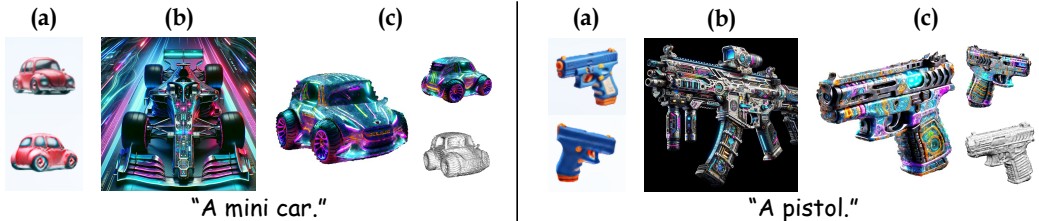

Figure 10: More samples of 3D object editing. (a) Coarse NeRF models. (b) Provided image prompts. (c) 3D objects generated by IPDreamer.

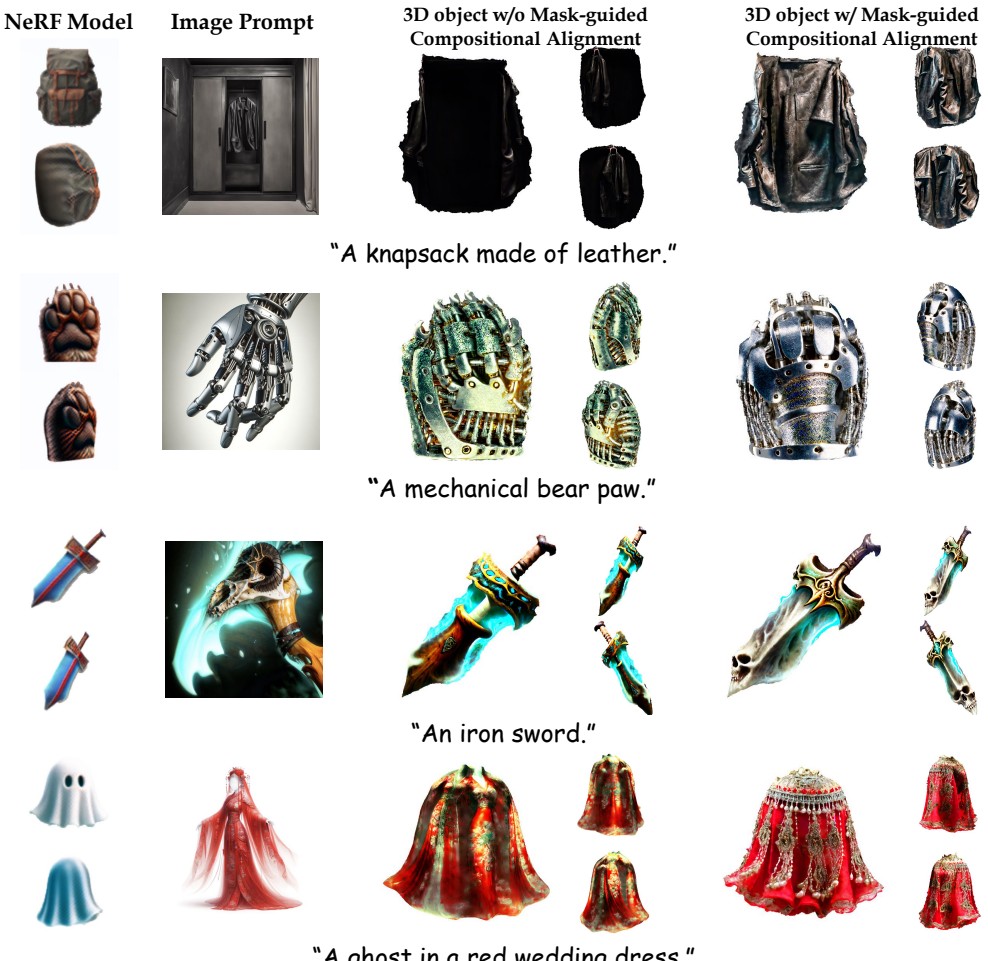

Figure 11: Comparison of 3D objects synthesis with and without Mask-guided Compositional Alignment.

by two distinct image prompts. As evident in Fig. 9, IPDreamer consistently produces impressive 3D object synthesis, regardless of the geometric shape of the acquired NeRF or the image prompts used for texture editing. The generated results highlight IPDreamer's powerful texture editing capabilities for 3D objects, suggesting its potential to serve effectively in the 3D gaming and video industries.

Besides, to further demonstrate the appearance guidance capability of IPSDS in generating 3D objects, we use two samples whose reference image prompts are particularly complex and somewhat different from the initial coarse NeRF models, as shown in Fig. 10. Even in such challenging cases, IPDreamer can still achieve high-quality 3D objects, such as the cyborg-style mini car generated in

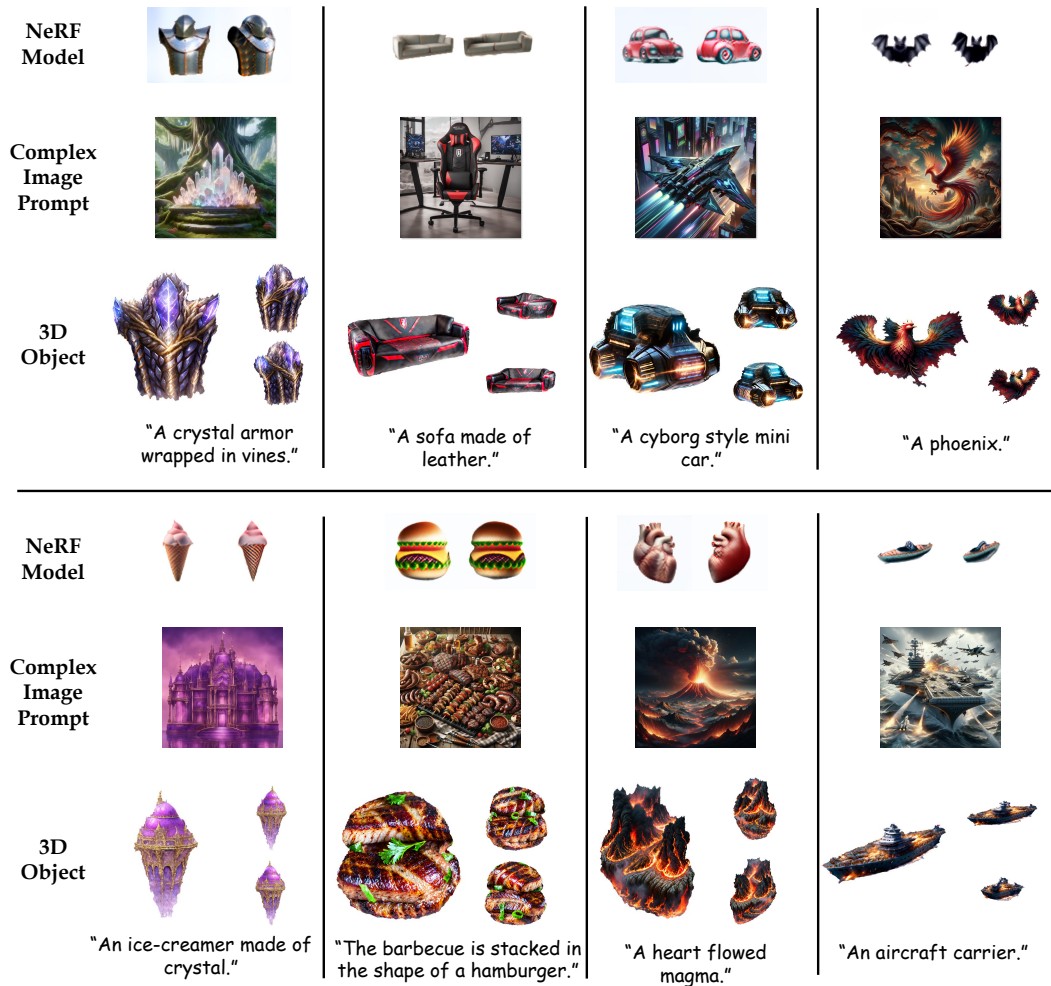

Figure 12: 3D objects synthesis with Mask-guided Compositional Alignment.

the first example, and the futuristic toy pistol in the second example. By utilizing IPDreamer's style editing ability for 3D objects, the generated results can be more diverse.

### A.1.2 MORE EXAMPLES OF 3D OBJECTS GUIDED WITH MASK-GUIDED COMPOSITIONAL ALIGNMENT

While IPDreamer can achieve remarkable 3D object synthesis in numerous challenging cases even without Mask-guided Compositional Alignment, difficulties emerge when the appearance of the supplied image prompts substantially diverge from the initial coarse NeRF model. To emphasize the potent 3D object optimization capability of Mask-guided Compositional Alignment within IP-Dreamer, we offer a comparison of the generated 3D objects with and without Mask-guided Compositional Alignment in Fig. 11. The outcomes validate the exceptional high-fidelity capability of Mask-guided Compositional Alignment. To further elucidate the superiority of Mask-guided Compositional Alignment, we present additional generation results in Fig. 12

### A.1.3 MORE TEXT-TO-3D GENERATION RESULTS

To further demonstrate the capability of our IPDreamer in generating the desired 3D objects, we provide additional qualitative comparison results in Fig. 13. The baseline methods, including ProlificDreamer (Wang et al., 2023) as well as single-image-to-3D methods (Hong et al., 2024; Tang et al., 2024; Shi et al., 2023a; Voleti et al., 2025), are also included. It is evident that for the ambiguous testing samples or lack a clear main subject, existing text-to-3D and single-image-to-3D

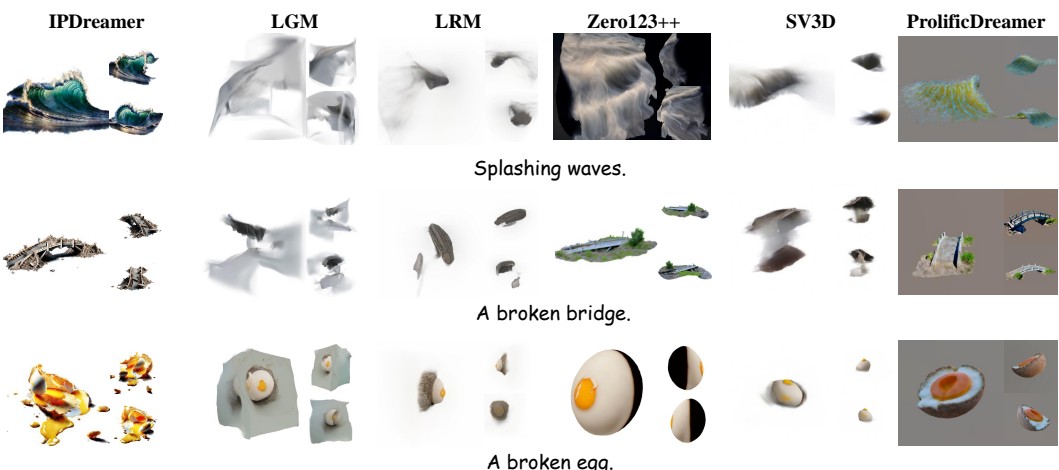

Figure 13: More Qualitative comparison of text-to-3D synthesis.

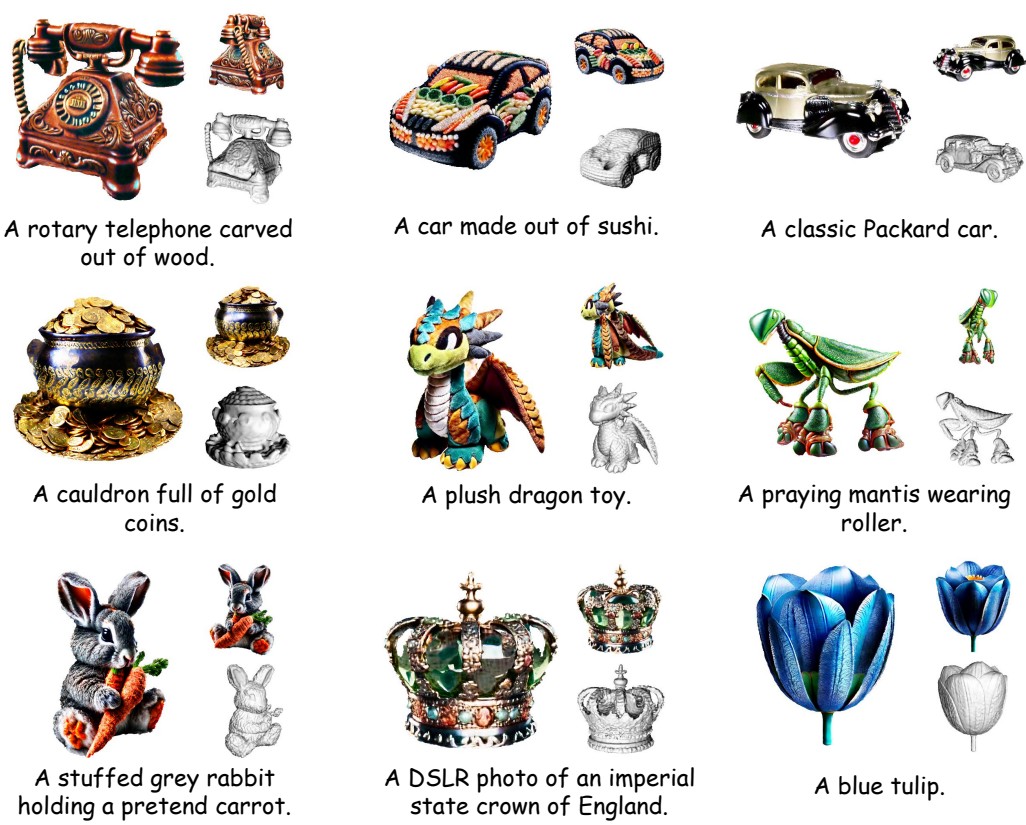

Figure 14: More generated 3D objects of IPDreamer.

methods struggle to produce the desired 3D results. In contrast, our IPDreamer succeeds in generating accurate 3D results. To further showcase the diversity of our model, we provide additional text-to-3D generation results in Fig. 14.

A praying mantis wearing roller.
Michelangelo-style statue of a dog reading news on a cellphone.
A matte painting of a castle made of cheesecake surrounded by a moat made of ice cream.
A chimpanzee dressed like Henry VIII king of England.
A 3D model of an adorable cottage with a thatched roof.
A plate piled high with chocolate chip cookies.
A vintage record player.
A car made out of cheese.
A beautifully carved wooden knight chess piece.
A car made out of sushi.
A squirrel-octopus hybrid.
A small saguaro cactus is planted in a clay pot.
A DSLR photo of an imperial state crown of England.
A rotary telephone carved out of wood.
A raccoon astronaut holding his helmet.
A classic Packard car.
A cauldron full of gold coins.
A blue tulip.
A stuffed grey rabbit holding a pretend carrot.
A plush dragon toy.
A broken egg.
A popped balloon.
Leaves flying in the wind.
A robot assembles itself.
Lightning.
The shining sun.
A melting ice cube.
Ripples on water.
A broken bridge.
Splashing waves.

Table 4: Textual prompts used in the quantitative comparison.

## A.2 IMPLEMENTAION DETAILS

### A.2.1 OPTIMIZATION

In this work, we conduct all of our experiments on one A100-SXM4-40GB GPU. In Stage 1, we optimize $5k$ steps with Adam optimizer Xie et al. (2022) to obtain a NeRF model. In Stage 2, we optimize $10k$ steps for geometry optimization and $15k$ steps for texture optimization. During each optimization progress in Stage 2, we initially sample the timesteps $t \sim \mathcal{U}(0.02, 0.98)$ for the first $5k$ steps, and then sample $t$ from $t \sim \mathcal{U}(0.02, 0.5)$ for the rest steps. Each optimization process in Stage 2 requires approximately 9GB GPU memory with batch size 1 and a rendering resolution of 512.

### A.2.2 TEXTUAL PROMPTS USED FOR COMPARISON

We provide the 30 randomly selected textual prompts for quantitative comparison and user study in Table. 4. To fully compare the generation capabilities of different methods and demonstrate the effectiveness of our method, the testing textual prompts include 20 textual prompts that are frequently used in previous text-to-3D methods as well as 10 relatively challenging textual prompts that do not have a clear main subject.

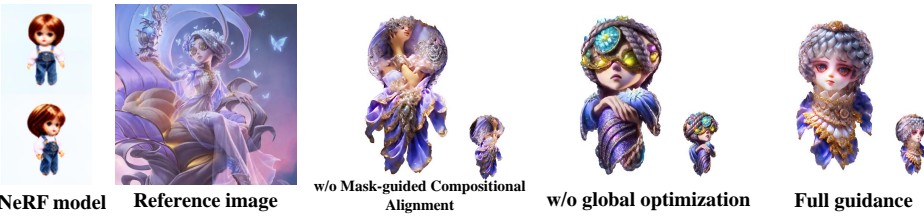

**NeRF model**  **Reference image**  **w/o Mask-guided Compositional Alignment**  **w/o global optimization**  **Full guidance**

Figure 15: Ablation study of global optimization.

### A.2.3 METRICS

We perform quantitative comparisons to evaluate IPDreamer's performance, with the following metrics:

- CLIP score Gal et al. (2022): We employ CLIP score in Section 4.2 of the main paper. By assessing the alignment between the textual descriptions and the rendered images of 3D objects from various viewpoints, we can judge whether text-to-3D methods successfully generate 3D objects that match the input textual prompts.

- Fréchet Inception Distance (FID) Heusel et al. (2017): To evaluate the quality of the generated results, we utilize FID to compare the similarity between the rendering images of 3D objects and the images generated by the text-to-image model, Stable Diffusion.

### A.2.4 USER STUDY

To further verify the quality of our generated results, we follow previous works Lin et al. (2023); Chen et al. (2023b); Wang et al. (2023) and conduct a user study by comparing IPDreamer with the six SOTA methods Poole et al. (2022); Lin et al. (2023); Chen et al. (2023b); Wang et al. (2023); Hong et al. (2024); Tang et al. (2024), under 16 prompts randomly selected from Table 4. Each of the 80 volunteers is provided with 16 pairs of results corresponding to the 16 prompts. In each pair, one from IPDreamer and one from a randomly selected baseline. Thus, there are a total of 1280 pairwise comparisons. The volunteers are then asked to choose the better result in terms of faithfulness, quality, and fidelity.

To enhance the reliability of our user study, we provide information on the demographic distribution of the participants. The 80 volunteers included 30 university students, 20 employees from internet companies, and 30 individuals without a computer science background. This diverse composition, encompassing both participants with relevant academic experience and those without, suggests that the results of our user study are reliable and generalizable.

### A.2.5 DETAILS OF GPT-4V ANALYSIS

In this section, we provide a detailed description of how GPT-4v is utilized to generate localization prompts. Specifically, the prompt given to GPT-4v is as follows:

*"You will act as an image analysis agent. Based on the input complex image condition <image>, you need to analyze which parts of the image can be used to guide 3D object synthesis. The multiview renderings of the initialized 3D model will be provided to you in the form of a video <video>. Based on the input image condition, you are required to generate textual prompts that describe the segmented partial images, which will be used to guide the segmentation of partial image features. Each segmented partial image must also have a corresponding localization prompt, mapping these partial image features onto the 3D object. Please respond in the following format: Partial image textual prompts: <text1>, <text2>, ... Corresponding localization textual prompts: <y1>, <y2>, ... Note that the numbering of the partial image textual prompts must correspond one-to-one with the localization textual prompts."*

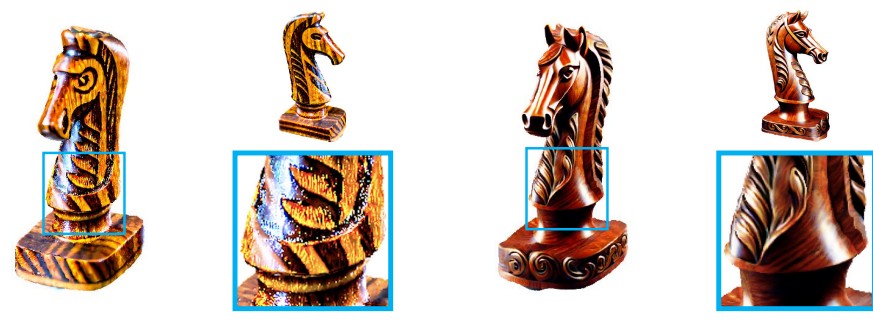

**w/o Image Prompt Enhancement**          **w/ Image Prompt Enhancement**

Figure 16: Ablation study of partial images enhancement.

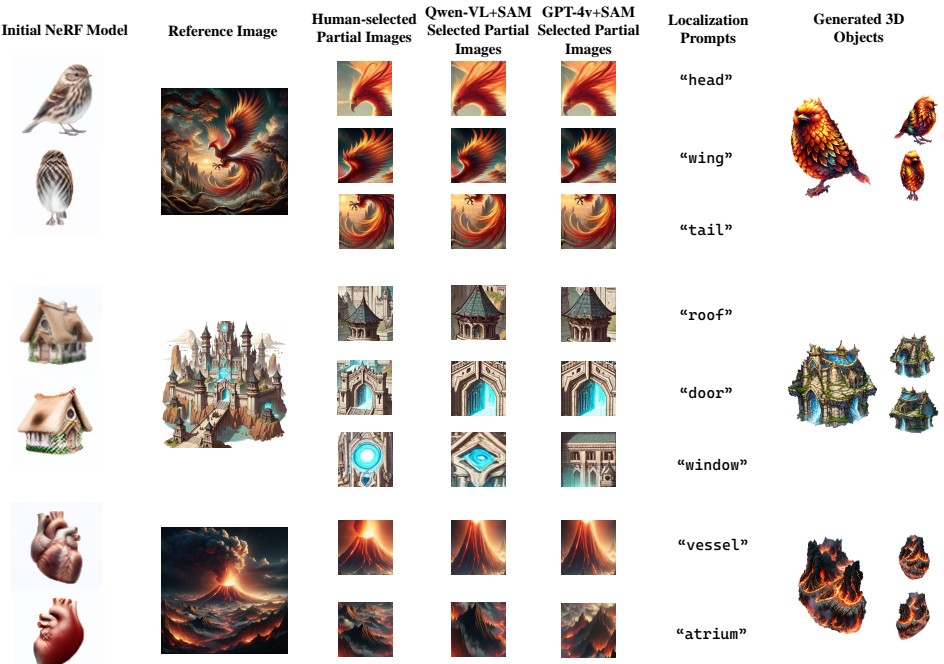

Figure 17: Ablation study of the accuracy of the GPT-4v and SAM.

## A.3 ADDITIONAL ANALYSIS

### A.3.1 EFFECTIVENESS OF GLOBAL OPTIMIZATION

To demonstrate the effectiveness of global optimization, We conduct an ablation study. As shown in Fig. 15, we present results optimized using only Mask-guided Compositional Alignment, only global optimization, and full guidance. When Mask-guided Compositional Alignment is omitted, the features of partial images fail to align properly with the corresponding regions on the 3D object. Conversely, without global optimization, the generated 3D object appears distorted and unnatural. By combining the Mask-guided Compositional Alignment strategy with global optimization, we can produce coherent, high-quality 3D objects. This highlights the role of Mask-guided Compositional Alignment in accurately localizing features and the importance of global optimization in enhancing the overall quality of the generated 3D object.

### A.3.2 EFFECTIVENESS OF PARTIAL IMAGES ENHANCEMENT

We utilize the super-resolution model to enhance the partial images in the main paper, especially when the resolution of partial images is low. In this part, We compare the generated 3D results with and without partial image enhancement. As shown in Fig. 16, the generated 3D models may exhibit noticeable noise and artifacts when the low-resolution partial images are not enhanced. In contrast, applying partial image enhancement leads to significantly improved 3D results, demonstrating the effectiveness of partial image enhancement.

### A.3.3 EFFECTIVENESS OF GPT-4V ANALYSIS

To validate the effectiveness of GPT-4v in analyzing and extracting parts, as well as to make the multi-image-guided 3D object generation process easier to understand, we provide additional visualizations in Fig. 17. These visualizations include manually extracted partial images, partial images obtained using GPT-4v and SAM, partial images obtained by Qwen-VL Bai et al. (2023) and SAM, and the 3D objects generated from the partial images obtained by GPT-4v and SAM. The results show that analyzing the input conditions and extracting partial images is not a particularly difficult task. The partial images extracted using MLLM (including GPT-4v and Qwen-VL) and SAM resemble those manually extracted. By leveraging analytical capabilities of GPT-4v and the Mask-guided Compositional Alignment strategy, our method can autonomously generate reasonable and high-quality 3D objects. This further demonstrates the effectiveness of GPT-4v and SAM in accurately obtaining partial images.

### A.4 FUTURE WORK

Our IPDreamer can leverage complex images to guide high-quality 3D object synthesis and editing. Compared to existing text-to-3D and single-image-to-3D methods, it enables a more flexible and controllable synthesis of the desired 3D results. In future work, we will focus on improving the alignment between the generated 3D objects and the provided complex image conditions, ensuring that the generated 3D results better reflect the finer details in the complex image.

Although our IPDreamer has successfully guided high-quality 3D object synthesis in most cases using complex images, there may still be cases of failure. Future work will focus on identifying and addressing such failure cases, while enhancing the generalization ability of IPDreamer and improving the quality of the generated 3D objects.

### A.5 SOCIAL IMPACT

Our IPDreamer does not have a direct negative impact on society. However, it is important to recognize the potential of high-quality 3D objects and ensure they are not adopted for malicious purposes.

