# OpenReview forum: "IPDreamer: Appearance-Controllable 3D Object Generation with Complex Image Prompts"
_ICLR.cc/2025/Conference — ICLR 2025 Poster_

### Official Review · Reviewer_4vLr · 2024-11-02

**Soundness:** 3
**Presentation:** 2
**Contribution:** 2
**Rating:** 5
**Confidence:** 4

**Summary:**

This paper present a novel method to capture intricate appearance features from the Image prompts, which is further used to enhance the alignment of the image prompts with the generated objects. Experiments demonstrate the proposed method generate objects which is well-aligned with image prompts, show better ability in complex generation.

**Strengths:**

* The paper proposes a novel framework for 3D generation by breaking an image prompt into several parts and adopting a multi-guidance optimization process. Experiments demonstrate the effectiveness of the proposed framework.
* The idea of the paper that breaks the complex images into several parts is interesting and good. Breaking a complex thing into parts makes a hard problem much easier.

**Weaknesses:**

* The written of the paper is not so clear, some details are lack:
  * The description on how to adopt GPT-4v to generate localization prompts is lack in the paper.
  * In Figure 1 (b), the author gives comparison between VSD and IPSDS on text-based generation. But is the proposed method IPSDS need an image prompt? How to compare IPSDS with VSD on text-based generation? Moreover, for the cases in Fig (a), could the author provide the images parts extracted from the reference image of the castles. It’s hard to understand how could we break such things into parts.
  * For eq.9 and eq.10,the author highlights that “they localize the features of the multiple images onto 3D object” in many places such as Line 321-322, 349-350, which makes me very confused. I think the author is adopting eq.9 and eq.10 to fuse information from different image parts to do SDS loss. Therefore, this description is inaccurate and leads to misunderstanding. 、
  * Some annotation in the equations are missing, like $Z$ in eq.9.
* In line 360, the author declares that a global optimization is further needed, which is achieved by simply concatenating all the features from the multiple images instead of adopting a mask based strategy. Why we need such a global optimization? What if we directly adopt global optimization without the mask-guided one? I think the author should provide such evaluation.
* Finally, I think the evaluation of the paper is not enough. The accuracy of adopting SAM and GPT-4v to break into parts is not evaluated. Moreover, I think the author should provide more visualization examples on the extracted image parts together with the generation results, which will make overall process easier to understand.

**Questions:**

See Weakness.

---

> ### Author Response · Authors · 2024-11-19
> **Response to Reviewer 4vLr (Part 1/2)**
>
> Thank you for your questions and suggestions. Below are the responses to the points you raised.
>
> **Q1: The written of the paper is not so clear, some details are lack:**
>
> > **Q1.1: The description on how to adopt GPT-4v to generate localization prompts is lack in the paper.**
>
> **A1.1:** Thank you for your feedback. In the revised version of the paper, we have added details on how GPT-4v is used to generate localization prompts in Appendix A.2.5. Specifically, the prompt provided to GPT-4v is as follows:
>
> _"You will act as an image analysis agent. Based on the input complex image condition < image >, you need to analyze which parts of the image can be used to guide 3D object synthesis. The multi-view renderings of the initialized 3D model will be provided to you in the form of a video < video >. Based on the input image condition, you are required to generate textual prompts that describe the segmented partial images, which will be used to guide the segmentation of partial image features. Each segmented partial image must also have a corresponding localization prompt, mapping these partial image features onto the 3D object. Please respond in the following format:_
>
> _Partial image textual prompts: < text1 >, < text2 >, ..._ <br>
> _Corresponding localization textual prompts: < y1 >, < y2 >, ..._
>
> _Note that the numbering of the partial image textual prompts must correspond one-to-one with the localization textual prompts."_
>
> > **Q1.2: In Figure 1 (b), the author gives comparison between VSD and IPSDS on text-based generation. But is the proposed method IPSDS need an image prompt? How to compare IPSDS with VSD on text-based generation? Moreover, for the cases in Fig (a), could the author provide the images parts extracted from the reference image of the castles. It’s hard to understand how could we break such things into parts.**
>
> **A1.2:** Thank you for your valuable questions. First, to clarify, Recent advancements in text-to-image generation have allowed single-image-to-3D methods to leverage these techniques for text-based 3D generation. Existing single-image-to-3D research has already showcased results in text-based 3D synthesis. However, it is crucial to note that the images generated by text-to-image models often lack distinct, well-defined foregrounds. This limitation implies that current single-image-to-3D methods struggle to consistently achieve reliable text-to-3D generation. In contrast, IPDreamer leverages complex image prompts to guide the 3D synthesis process, allowing it to produce high-quality, stable results for text-to-3D tasks.
>
> Regarding Fig. 1(a), it is important to clarify that Mask-guided Compositional Alignment combined with IPSDS is necessary only when there are substantial appearance and semantic differences between the initialized coarse 3D object and the reference image, or when multiple complex images are used to guide the optimization of a single 3D object. For the case in Fig. 1(a), high-quality 3D object generation can be achieved using only equations (2) to (7), without the need to extract partial images from the reference complex images. This further demonstrates the effectiveness of our IPSDS.
>
> > **Q1.3: For eq.9 and eq.10,the author highlights that “they localize the features of the multiple images onto 3D object” in many places such as Line 321-322, 349-350, which makes me very confused. I think the author is adopting eq.9 and eq.10 to fuse information from different image parts to do SDS loss. Therefore, this description is inaccurate and leads to misunderstanding.**
>
> **A1.3:** Your understanding is correct. Equations (9) and (10) modify equation (8) when there is a significant difference between the initialized coarse 3D object and the reference image, or when there are multiple complex images served as conditions. As you rightly pointed out, equations (9) and (10) are used to adjust the cross-attention mechanism during the SDS loss calculation process, which enables the localization of features from multiple images onto the 3D object. Based on your feedback, we have updated the description in the revised paper for greater clarity. If any points remain unclear, please feel free to ask.
>
> > **Q1.4: Some annotation in the equations are missing, like Z in eq.9.**
>
> **A1.4:** Thank you for pointing this out. We appreciate your careful review. The variable Z in Equation (9) is defined earlier in the text (see line 283) as the query features. Additionally, we have carefully reviewed the paper and added missing annotations to ensure that all variables are clearly described in the revised paper.

---

> > ### Author Response · Authors · 2024-11-19
> > **Response to Reviewer 4vLr (Part 2/2)**
> >
> > **Q2: In line 360, the author declares that a global optimization is further needed, which is achieved by simply concatenating all the features from the multiple images instead of adopting a mask based strategy. Why we need such a global optimization? What if we directly adopt global optimization without the mask-guided one? I think the author should provide such evaluation.**
> >
> > **A2:** Thank you for your valuable suggestion. In the revised paper, we have conducted comprehensive ablation studies (now included in Appendix A.3.1) to evaluate the individual and combined effects of these optimization strategies.
> >
> > Our experimental results demonstrate that:
> > 1. Without mask-guided alignment, precise image feature localization fails;
> > 2. Without global optimization, the generated 3D objects exhibit significant artifacts;
> > 3. The combination of both components yields optimal results, producing high-quality 3D objects.
> >
> > These findings validate both the effectiveness of mask-guided alignment for feature localization and the necessity of global optimization for overall quality enhancement.
> >
> > **Q3: Finally, I think the evaluation of the paper is not enough. The accuracy of adopting SAM and GPT-4v to break into parts is not evaluated. Moreover, I think the author should provide more visualization examples on the extracted image parts together with the generation results, which will make overall process easier to understand.**
> >
> > **A3:** Thank you for your suggestion. To further demonstrate the effectiveness of GPT-4v in analyzing and extracting parts, as well as to make the multi-image-guided 3D object generation process easier to understand, we have added additional visualizations in Appendix A.3.3 of the revised paper. These visualizations include partial images obtained using GPT-4v and SAM, manually extracted partial images, and the 3D objects generated using the partial images obtained by GPT-4v and SAM. The results show that the partial images extracted using GPT-4v and SAM closely resemble those manually extracted, and the generated 3D objects are both reasonable and of high quality. This further underscores the effectiveness of GPT-4v and SAM in obtaining accurate partial images. Moreover, it is worth noting that the key to achieving high-quality 3D object optimization guided by multiple images lies in the Mask-guided Compositional Alignment strategy.

---

> > > ### Comment · Reviewer_4vLr · 2024-11-27
> > >
> > > I thank authors for providing those revisions and replies towards my concerns.  The authors' reply partly address my concerns. But I still have concerns about the generalization ability of this method. The method builds on a very strong assumption that  the input image could be divided into reasonable semantic regions which is not always the case. Therefore, I tend to maintain my score.

---

> ### Author Response · Authors · 2024-11-27
> **Response to Reviewer 4vLr**
>
> Dear Reviewer 4vLr,
>
> We sincerely thank you for your constructive feedback and insightful comments. The core contribution of this paper is the ability to use single or multiple complex images to guide 3D object synthesis, enabling high-quality texture editing and generation without a clear main subject. Specifically, IPSDS and Mask-guided Compositional Alignment are the key contributions of our work, and the use of MLLM analysis serves to further refine the method.
>
> Regarding your concern that "the input image cannot be divided into reasonable semantic regions," we have not encountered any issues in our experiments where IPDreamer was unable to effectively guide the 3D results using complex images. Besides, the ability to guide 3D object synthesis with multiple complex images represents a significant improvement over the generalization ability of current single-image-to-3D methods. While we have not encountered unsatisfactory results in texture editing so far, we acknowledge that your consideration is valid. In response, we have added the following statement in the Future Work section (Appendix A.4) of the revised paper: "Although our IPDreamer has successfully guided high-quality 3D object synthesis in most cases using complex images, there may still be cases of failure. Future work will focus on identifying and addressing such failure cases, while enhancing the generalization ability of IPDreamer and improving the quality of the generated 3D objects."
>
> Once again, thank you for your valuable suggestions, which have greatly helped us refine the paper.
>
> Sincerely,  \
> The Authors

---

> > ### Author Response · Authors · 2024-11-30
> > **Gentle Follow-Up**
> >
> > Dear Reviewer 4vLr,
> >
> > Once again, we thank you for your insightful feedback. This comment is a follow-up to the previous one. The core idea of our paper is to extract features from complex images and localize these features onto the 3D objects, enabling flexible and controllable high-quality 3D object generation. Therefore, IPSDS and Mask-guided Compositional Alignment are key to allowing our IPDreamer to use single or multiple complex images as conditional inputs for 3D synthesis.
> >
> > Regarding the segmentation of complex images into partial images using MLLM+SAM, the requirements for partial images are actually not very strict; we only need a rough segmentation of the relevant regions. As demonstrated in the paper, by leveraging the powerful capabilities of IPSDS and Mask-guided Compositional Alignment, IPDreamer can stably generate high-quality 3D objects across a variety of scenarios.
> >
> > We sincerely appreciate your constructive input, and if you have any further questions or concerns, we would be happy to address them.
> >
> > Sincerely,  \
> > The Authors

---

### Official Review · Reviewer_sW6c · 2024-11-03

**Soundness:** 3
**Presentation:** 2
**Contribution:** 3
**Rating:** 6
**Confidence:** 2

**Summary:**

The paper introduced a controllable 3D object generation approach using image prompts (similar to style transfer). The proposed IPDreamer approach is a novel method that could capture the intricacies of appearance features from image prompts, and could generate high fidelity and controllable 3D objects. The approach is tested on some public benchmarks with user studies available as well, and was proven to be effective.

**Strengths:**

- The paper is tackling an important and very challenging 3D genAI problem. Comparing to existing approaches, the IPDreamer could edit the objects using more complex image prompts
- The introduced prompt score distillation sampling approach is a reasonable formulation that builds on existing SDS approaches, and the masked-guided alignment strategy seems to be highly effective
- Experimental results suggest that the approach is better comparing to other counterparts. User studies is also provided.

**Weaknesses:**

I think this is a nice paper and a good extension to many of the existing approaches. The final output of the algorithm seems to be good enough. I do have a few clarification questions that I hope the authors could address in future revisions of the papers:
- The paper leverages GPT-4v as MLLM inputs. How accurate should the MLLM be, in case people don't have access to this advanced MLLM algorithm? Would the output become much worse?
- It's very nice to conduct user studies for genAI works in general. Could authors provide more demographics information in the appendix section? (age, gender, background, etc)
- I don't fully understand Fig 1b, especially the right images -- what is the contents in the input and what is the actual real-world application of this particular input/output pair?

**Questions:**

See the weakness section

---

> ### Author Response · Authors · 2024-11-19
> **Response to Reviewer sW6c**
>
> Thank you for your thoughtful review and support. We address each of your questions in detail below.
>
> **Q1: The paper leverages GPT-4v as MLLM inputs. How accurate should the MLLM be, in case people don't have access to this advanced MLLM algorithm? Would the output become much worse?**
>
> **A1:** The incorporation of MLLM serves to automate the analysis of reference complex images and 3D objects, specifically to:
> 1. Identify regions requiring segmentation in reference images;
> 2. Generate appropriate localization prompts for positioning these segments on the 3D object.
>
> As detailed in Appendix A.3.3 of the revised paper, we present examples of partial images, corresponding localization textual prompts, and optimized 3D objects generated through the addition of Mask-guided Compositional Alignment during the IPSDS optimization process. These results demonstrate that analyzing input conditions is not particularly difficult, and partial images and localization prompts can be manually obtained.
>
> To demonstrate the robustness of our approach without relying solely on GPT-4v, the revised paper includes partial images obtained using the open-source Qwen-VL model combined with SAM. The partial images generated by Qwen-VL and SAM were comparable to those produced by GPT-4v and SAM, showing that even without access to GPT-4v, reasonable analysis results can still be achieved using alternative MLLMs such as Qwen-VL.
>
> By the way, it is worth noting that the generation of high-quality 3D objects guided by multiple complex images relies more on the effectiveness of the Mask-guided Compositional Alignment strategy.
>
>
> **Q2: It's very nice to conduct user studies for genAI works in general. Could authors provide more demographics information in the appendix section? (age, gender, background, etc)**
>
> **A2:** Thank you for your thoughtful question. While we must maintain participant privacy by withholding specific personal details such as age and gender, we can share the professional composition of our study participants to demonstrate the diversity of our sample:
>
> Our 80 volunteers comprised:
> - 30 university students
> - 20 internet company professionals
> - 30 participants from non-computer science backgrounds
>
> This balanced distribution of participants, including both those with and without technical expertise, helps establish the reliability and generalizability of our user study results. We have included these demographic details in Appendix A.2.4 of the revised manuscript.
>
>
> **Q3: I don't fully understand Fig 1b, especially the right images -- what is the contents in the input and what is the actual real-world application of this particular input/output pair?**
>
> **A3:** Thank you for your question. First, I would like to clarify that Fig. 1(b) displays 2D rendered results of 3D objects. Additionally, in the revised version of the paper, we have included the results of single-image-to-3D generation in Fig. 1(b). The purpose of this figure is to demonstrate that existing text-to-3D and single-image-to-3D methods fail to generate reasonable results for cases where the subject is unclear, such as "Leaves flying in the wind" or "Ripples on the water." In contrast, our method can leverage complex images to guide 3D object synthesis, enabling the generation of rational, high-quality 3D objects even for these challenging examples.
>
> In industrial scenarios, 3D modeling is often applied to create special effects for complex scenes, such as "Leaves flying in the wind." These effects typically do not have a distinct physical entity, making it difficult for text-to-3D and single-image-to-3D methods to generate accurate results. Our method can handle such cases, demonstrating its practical value in producing realistic and high-quality scene effects.

---

> > ### Comment · Reviewer_sW6c · 2024-11-26
> > **reply**
> >
> > I thank authors for providing those revisions and addressing my concerns. After reading other reviewers' questions, it looks like my concerns were not just from myself, and it'd be good if authors could include those in future revisions of the paper. Regarding ratings, I think mine stands unchanged, but some of the concerns from other reviewers (such as citing related works, making certain descriptions clearer, etc, are all valid)

---

> ### Author Response · Authors · 2024-11-27
> **Thanks for your response**
>
> Dear Reviewer sW6c,
>
> We sincerely thank you for your suggestions. Your feedback has been instrumental in helping us improve our paper. In the revised paper, we have made modifications based on the recommendations provided by all reviewers. Additionally, we plan to submit a second version of the revised paper before the revised paper submission deadline. In the new version, we will include more text-to-3D generation results based on the reviewers' new suggestions. If you have any further suggestions, please feel free to share them with us—we greatly appreciate your valuable input.
>
> Once again, thank you for your patience in reviewing our work and for providing constructive feedback.
>
> Sincerely, \
> The Authors

---

### Official Review · Reviewer_zssc · 2024-11-03

**Soundness:** 1
**Presentation:** 2
**Contribution:** 2
**Rating:** 3
**Confidence:** 3

**Summary:**

The paper introduces a text/image-to-3D approach for controlling the appearance of generated 3D objects given complex input images where the subject is not clearly identified.
The proposed approach encompasses multiple components.
First, IPAdapter image encoder is used to extract image features that are used as texture guidance within the Score Distillation Sampling (SDS).
To be able to handle complex images with multiple components, a mask-guided compositional alignment strategy exploits a Multi-Modal Language Model (MLLM) to provide localization part labels given the image and the provided coarse Nerf model.
Then, cross-attention maps are used to localize those parts by computing attention between the image feature and the textual labels produced by the MLLM.
Finally, the localized parts are optimized jointly to produce a globally consistent 3D object.
Experiments show that the proposed approach produces high-quality results that abide by the guidance image.

**Strengths:**

- The idea of splitting complex objects into parts that are optimized jointly is interesting and can be potentially employed for more complicated 3D scenes.

- The method section is comprehensive and provides an overview of SDS, making it self-contained.

- The visual quality of the provided results is compelling.

**Weaknesses:**

- The paper primarily focuses on controlling the generation of 3D objects from complex input images. As noted in line 537, "IPDreamer addresses the limitations of existing text-to-3D and **single-image-to-3D** methods." However, the paper does not include comparisons with relevant single-image-to-3D methods, such as [1] and [2]. Could the authors clarify why these comparisons were omitted?

- In Figure 7, the qualitative comparison presents different samples for each method. Conventionally, all methods are evaluated on the same samples to ensure consistency in comparisons. Could the authors provide insight into this choice?

- The proposed method incorporates several additional components beyond the standard SDS pipeline, including ChatGPT, SAM, ControlNet, and IPAdapter. Could the authors provide details on the runtime overhead introduced by each component, as well as the overall runtime?

- The method illustration in Figure 2 appears challenging to interpret. It does not effectively aid in understanding the proposed pipeline, and I found it difficult to correlate it with the text. A more intuitive figure might improve readability and clarity.

[1] Shi, Ruoxi, et al. "Zero123++: a single image to consistent multi-view diffusion base model." arXiv preprint arXiv:2310.15110 (2023).

[2] Voleti, Vikram, et al. "Sv3d: Novel multi-view synthesis and 3d generation from a single image using latent video diffusion." European Conference on Computer Vision. Springer, Cham, 2025.

**Questions:**

- I do not understand Figure 1b. What is being generated, a 3D shape or an image? both the leaves and the water ripples look like images!

- What is the difference between equations (11-13) and (14-17)? Are both used during optimization?

- What is the impact of employing the super-resolution model, ControlNet tiling, on the final generated quality?

---

> ### Author Response · Authors · 2024-11-19
> **Response to Reviewer zssc (Part 1/2)**
>
> Thank you for your patience and for taking the time to review our paper. Below, we respond to the points you raised.
>
> **Q1: The paper primarily focuses on controlling the generation of 3D objects from complex input images. As noted in line 537, "IPDreamer addresses the limitations of existing text-to-3D and single-image-to-3D methods." However, the paper does not include comparisons with relevant single-image-to-3D methods, such as Zero123++ and SV3D. Could the authors clarify why these comparisons were omitted?**
>
> **A1:** Thank you for your question. We would like to clarify that both LRM and LGM, which **were already included in our comparisons**, are single-image-to-3D generation methods. To avoid any confusion for future readers, we have now explicitly stated in the revised paper that LRM and LGM are single-image-to-3D methods. Additionally, in response to your suggestion, we add comparisons with Zero123++ and SV3D. The experimental results show that our method still achieves the highest FID and CLIP scores.
>
> **Q2: In Figure 7, the qualitative comparison presents different samples for each method. Conventionally, all methods are evaluated on the same samples to ensure consistency in comparisons. Could the authors provide insight into this choice?**
>
> **A2:** The reason we presented the qualitative comparison in this manner was to showcase a broader range of examples within the limited space of the paper. This allowed us to better demonstrate the versatility and superior performance of our method across diverse scenarios. Additionally, in the revised paper, we have updated Fig. 7 to show comparisons of our method with all other methods using the same sample. Based on the experimental results, we would like to point out that single-image-to-3D methods rely on clear foreground subjects as input, however such images with distinct subjects are relatively difficult to obtain. For example, objects like "the shining sun," which emit rays of light, are challenging for both single-image-to-3D and text-to-3D methods to generate effectively. In contrast, our IPDreamer can generate them much more accurately.
>
> **Q3: The proposed method incorporates several additional components beyond the standard SDS pipeline, including ChatGPT, SAM, ControlNet, and IPAdapter. Could the authors provide details on the runtime overhead introduced by each component, as well as the overall runtime?**
>
> **A3:** The runtime for running ChatGPT, SAM, and ControlNet is around 3-4 minutes, while the optimization time for the 3D object is approximately 1 hour and 20 minutes. Therefore, the overall optimization time is comparable to other methods that generate high-quality 3D results, such as Fantasia3D and MVDream. While our method isn't the fastest, it is on par with the speed of Fantasia3D and faster than ProlificDreamer, which also generates high-quality results. The relative computational speed ratio of our method, Fantasia3D, and ProlificDreamer is approximately 1:1:1.5. Although real-time performance remains a challenge for current high-quality 3D generation methods, our approach, leveraging complex image prompts, enables the production of high-quality 3D assets, making it a practical solution for detailed and controllable 3D generation.
>
> **Q4: The method illustration in Figure 2 appears challenging to interpret. It does not effectively aid in understanding the proposed pipeline, and I found it difficult to correlate it with the text. A more intuitive figure might improve readability and clarity.**
>
> **A4:** Thank you for your suggestion. The original version of Fig. 2 was designed to highlight the core contributions of our paper. In response to your feedback, we have updated Fig. 2 in the revised paper to present a clearer and more intuitive depiction of the full IPDreamer pipeline. In the new framework, we illustrate the complete generation process. If you have any further questions about the pipeline, please feel free to raise them.

---

> ### Author Response · Authors · 2024-11-19
> **Response to Reviewer zssc (Part 2/2)**
>
> **Q5: I do not understand Figure 1b. What is being generated, a 3D shape or an image? both the leaves and the water ripples look like images!**
>
> **A5:** Thank you for your question. The results shown in Fig. 1(b) are the rendered results of 3D objects. In the original version of our paper, the background of rendered results in Fig. 1(b) was not set to white. We have updated the Fig. 1(b) in the revised paper. We also appreciate your recognition of the quality of the generated 3D results in this challenging example.
>
> **Q6: What is the difference between equations (11-13) and (14-17)? Are both used during optimization?**
>
> **A6:** Thank you for your question. When there are multiple input images, the optimization process consists of two key stages: First, equations (11-13) implement Mask-guided Compositional Alignment to localize features from multiple images onto specific regions of the 3D object. Subsequently, equations (14-17) perform global optimization based on IPSDS to enhance the overall quality and coherence of the generated 3D result.
>
> To demonstrate the necessity of both components, we have included comprehensive ablation studies in Appendix A.3.1, comparing results generated under three conditions:
> 1. Using only Mask-guided Compositional Alignment
> 2. Using only global optimization
> 3. Using our full guidance approach
>
> The results clearly show that without Mask-guided Compositional Alignment, features from input images fail to properly localize on the 3D object. Conversely, without global optimization, the generated 3D objects exhibit notable artifacts. These comparisons validate the essential role of both components in achieving high-quality 3D synthesis.
>
> **Q7: What is the impact of employing the super-resolution model, ControlNet tiling, on the final generated quality?**
>
> **A7:** Thank you for your question. The super-resolution model is used in our paper to enhance the partial image prompts, especially when the resolution of these images is too low. In Appendix A.3.2 of the revised paper, we compare the 3D results generated with and without partial image enhancement. As shown in the visual comparison, using low-resolution partial images as prompts can lead to 3D results that contain unnatural noise. In contrast, the enhanced partial images result in significantly better 3D outputs, demonstrating the effectiveness of employing super-resolution to improve the quality of the final generated results.

---

> ### Comment · Reviewer_zssc · 2024-11-25
>
> Thank you for taking the time to respond to my questions and comments!
> I still have several concerns about the paper.
>
> 1- As noted by `PRnJ`, the generated 3D objects do not align with the input image (Figure 1, 5 ). This is a fundamental requirement for image-to-3D approaches, which is not fulfilled by your method. For artists, it is crucial that the generated 3D assets align well with their input images.
>
> 2- I understand that your approach is inspired by IPAdapter in Text-to-Image diffusion models, but I am still wondering how it could be useful in the 3D generation domain!
> In text-to-3D, the input is text, and the model is free to generate any style, while for image-to-3d, it needs to abide by the input image.
> Your approach seems to be in the middle between the two cases, making it difficult to position and compare against existing approaches for 3D generation.
>
> 3- I still have concerns about using different samples for different methods in Figure 7. To be able to position your approach amongst existing approaches, the same samples should be used for all comparisons. "The shinning sun" example that you showed at the bottom of the figure is not sufficient to judge.
>
> Minor:
> a- The caption of table 1 and 2 say "text-to-3D", but both Zero123 and SV3D are "image-to-3d".

---

> > ### Author Response · Authors · 2024-11-25
> > **Response to Reviewer zssc**
> >
> > Dear Reviewer zssc,
> >
> > We sincerely appreciate your feedback and thoughtful comments. Below are our responses to your additional concerns:
> >
> > **Response 1:**
> > Regarding Fig. 1(a) and Fig. 5, we would like to emphasize that these tasks are meant to demonstrate the 3D object texture editing capability of our proposed method. In industrial applications, such as 3D games, it is common practice to start with a basic 3D object and then allow 3D designers to refine it based on a reference poster, which is often a complex image. Existing text-to-3D and single-image-to-3D methods cannot utilize complex images as conditions to achieve high-quality 3D object texture editing.
> >
> > Our Fig. 1(a) and Fig. 5 demonstrate that our method can align the appearance of a given 3D object with a complex image while preserving the semantic and structural properties of the original 3D object. This proves that our approach is capable of achieving high-quality 3D texture editing, showcasing the potential of IPDreamer for practical deployment in industrial applications and 3D asset creation.
> >
> > **Response 2:**
> > To address this question, we would like to reiterate the limitations of existing text-to-3D and single-image-to-3D methods. While text-to-3D methods can generate high-quality 3D objects, their results are often uncontrolled in terms of appearance. On the other hand, single-image-to-3D methods typically require the input image to have a clear subject, which is often challenging to obtain.
> >
> > Our approach overcomes these limitations by leveraging complex images to guide 3D synthesis. It enables stable and controllable generation of high-quality 3D objects, making it suitable for both 3D object texture editing and synthesizing 3D objects with unclear edges. When generating 3D objects with specific appearances that text-to-3D and single-image-to-3D methods cannot achieve, our method can utilize complex images to guide the desired 3D object generation.
> >
> > **Response 3:**
> > Due to the many comparison methods (eight in total) and the need to demonstrate the diverse capabilities of our approach within the limited space available, we opted for the current presentation format to showcase the generative results. That said, your suggestion is very valuable, and we are considering ways to further improve and more comprehensively demonstrate the effectiveness of our approach in the future.
> >
> > **Response to the Minor Issue:**
> > Existing single-image-to-3D methods can be combined with text-to-image models to perform the text-to-3D generation task. Related single-image-to-3D papers also demonstrate such results. However, text-to-image generation methods often fail to produce clear subjects in the images, while our method can utilize such images with unclear subjects to guide 3D synthesis. This is why the "sun" generated by our method outperforms that of the compared single-image-to-3D methods, further highlighting our method’s ability to achieve stable and controllable high-quality text-to-3D synthesis.
> >
> > Thank you again for your insightful feedback and for raising these important questions.
> >
> > Sincerely, \
> > The Authors

---

> > > ### Author Response · Authors · 2024-11-30
> > > **Gentle Follow-Up**
> > >
> > > Dear Reviewer zssc,
> > >
> > > We appreciate the insightful feedback you provided. In response to your suggestions, we have included additional comparisons of text-to-3D generation tasks in Appendix A.1.3 of the revised paper. The updated results not only demonstrate the richness and stability of the 3D objects generated by IPDreamer, but also show that our method produces more desirable 3D outputs compared to existing text-to-3D and single-image-to-3D approaches.
> > >
> > > Additionally, we have included a more detailed global letter outlining the contributions and applications of IPDreamer. We would be grateful if you could let us know whether our revisions have addressed your concerns or if you have any further questions.
> > >
> > > Sincerely,   \
> > > The Authors

---

> > > ### Comment · Reviewer_zssc · 2024-12-01
> > >
> > > I completely agree with the limitations you mentioned in `Response 2`:
> > > - In text-to-3D approaches, the user cannot fully control the appearance.
> > > - In existing image-to-3D approaches, the user cannot utilize complex images effectively.
> > >
> > > Therefore, the first natural solution is what you mentioned in the general response above:
> > >
> > > "
> > > The core idea of our paper is to **extract corresponding features from complex images and align them with 3D objects**.
> > > "
> > >
> > > Let’s take the first row of Figure 1 as an example and see if this objective is achieved. Both the coarse NeRF and the reference images include towers. Naturally, I would expect your approach to map the tower's style from the reference image to the 3D shape. However, this was not achieved, and what happened is that only the overall style of the reference image is transferred, with no intelligent mapping of details—essentially the same as IP-Adapters in text-to-image diffusion models.
> > >
> > > I believe the focus of the paper needs to be refined to make the object more clear and to deliver results that match this objective.
> > > Also, the qualitative comparisons need to be done systematically in a way that facilitates judging the quality of the proposed approach compared to existing methods.
> > > Finally, with the emergence of Large Reconstruction Models that offer very fast generation times, your method falls behind in terms of efficiency, making it less attractive to build upon.
> > >
> > > Therefore, I am maintaining my original score.

---

> ### Author Response · Authors · 2024-12-01
> **Response to Reviewer zssc**
>
> Dear Reviewer zssc,
>
> Thank you for your feedback. First, we would like to clarify that Figure 1 is not simply a style transfer; it adjusts the appearance based on the shape of the initial 3D object. Please take another careful look at the texture editing results presented in the paper (Fig. 1, Fig. 5, Fig. 6, Fig. 9, Fig. 10, Fig. 11, Fig. 12, etc.). Each result is not just a style transfer but aligns the features of the input complex image with the shape and semantics of the initial 3D object, generating high-quality and rational results while preserving the shape and semantics of the initial object. Moreover, for geometric details, as shown in Fig. 8, our IPSDS-geo also fine-tunes the geometric details, which is definitely not just a style transfer.
>
> At the same time, the Large Reconstruction Models that you mentioned, which offer very fast generation times, are typically trained on datasets like Objaverse, which consist of clean, single-object 3D models. Generating complex scenes, such as "leaves flying in the wind," is still a challenging task for these methods. Therefore, existing fast-generation methods struggle with the synthesis of complex, high-quality 3D objects.
>
> We sincerely appreciate your feedback and fully respect your judgment, but please allow us to express that overlooking the key contributions of the paper would also be unreasonable. Thank you for helping us refine the paper, and we wish you all the best.
>
> Sincerely,    \
> The Authors

---

> > ### Author Response · Authors · 2024-12-02
> > **Gentle Follow-up**
> >
> > Dear Reviewer zssc,
> >
> > Apologies for the additional message, but we would like to provide some further clarification in response to your previous comments. After reviewing your feedback more carefully, we believe that you may have concerns regarding the value of the texture-editing presented in this paper. For this point, we encourage you to refer to the application section in the global letter, where we provide a variety of application scenarios. In these scenarios, it is crucial that the geometry of the initial 3D object does not undergo significant changes. If you are curious about the specifics of these applications, we would be happy to provide further details.
> >
> > Meanwhile, regarding texture-editing, if you are aware of any existing methods, similar to ours, that can use complex images to achieve high-quality 3D results that align with the complex image while maintaining the structure of the 3D object, and even reaching a commercial-quality level, we would appreciate it if you could share them here. **If no such methods exist, rejecting a work that holds practical applicability and lacks comparable alternatives seems a bit unusual.** Of course, we fully respect your judgment.
> >
> > Once again, we would like to express our sincere gratitude and respect for your valuable feedback. We truly appreciate your willingness to engage with us and contribute to refining this paper. If you have any further questions, we welcome them and will address them thoroughly.
> >
> > Sincerely,  \
> > The Authors

---

### Official Review · Reviewer_PRnJ · 2024-11-04

**Soundness:** 2
**Presentation:** 2
**Contribution:** 2
**Rating:** 6
**Confidence:** 4

**Summary:**

The paper introduces IPDreamer which, by leveraging the complex image prompts for the first time, can generate detailed 3D objects. To achieve this task, IPDreamer first proposes an Image Prompt Score Distillation Sampling (IPSDS) that leverages both RGB features and normal features to help guide the denoising process. The authors further introduce a mask-guided compositional alignment strategy that allows for extracting corresponding features from different images of the same objects, further improving the details of the 3D generation. Extensive qualitative and quantitative experiments have been provided in the paper.

**Strengths:**

+ The paper is the first time to consider generating 3D objects from complex images. It's quite interesting considering the current progress of the current 2D generative models.

+ The paper is well-written and easy to follow.

**Weaknesses:**

- Fig.1 is not clear. It's not able to showcase that existing methods struggle with complex images.

- The results showcased are not quite aligned with the input image.

- The masks in Fig.4 are not quite aligned with the corresponding parts.

- It's hard to see the effectiveness of mask-guided compositional alignment.

- The results provided in Fig. 5 are not very good.

- What if we apply the best text-to-2D diffusion model to the DreamFusion or other text-to-3D pipeline and carefully design the text prompts? For example, the text-to-2D diffusion model that's capable of generating complex and high-resolution images.

**Questions:**

Based on my comments on the strengths and weaknesses, I currently still lean a little bit toward the positive rating. I would like to hear from the other reviewers and the authors during the rebuttal.

---

> ### Author Response · Authors · 2024-11-19
> **Response to Reviewer PRnJ (Part 1/2)**
>
> Thank you for your thoughtful review and recognition of our research contributions. We now provide clarifications for the points you raised:
>
> **Q1: Fig.1 is not clear. It's not able to showcase that existing methods struggle with complex images.**
>
> **A1:** Thank you for your suggestion. We have updated Fig. 1(b) to include the generated results from the single-image-to-3D methods, along with the corresponding input image. It is now clearer that, for the input image with unclear subject, the single-image-to-3D methods cannot generate rational 3D results. To provide context for our observation about existing single-image-to-3D methods' limitations with complex images: these approaches are typically trained on datasets like Objaverse, which are characterized by clean, single-object 3D models. Consequently, they are inherently optimized for processing images with well-defined, isolated foreground objects. This fundamental constraint explains their limited performance when handling complex images characterized by rich visual content and sophisticated compositional elements, as detailed in our manuscript.
>
> **Q2: The results showcased are not quite aligned with the input image.**
>
> **A2:** Thank you for your question. IPDreamer's core innovation lies in balancing two objectives: preserving the structural integrity of the initial 3D object while incorporating visual elements from complex input images. While the provided coarse 3D objects may be semantically similar to the input complex images, they differ in structural morphology. Our method aligns the appearance of the generated 3D objects with the input images, while maintaining the structure and semantics of the coarse 3D objects as much as possible. This is why our results are not "quite" perfectly aligned with the input image.
>
> Our method makes two key contributions:
> 1. Generating meaningful 3D objects from complex images with ambiguous boundaries;
> 2. Performing high-quality texture editing on coarse 3D objects using complex image references.
>
> This design is particularly valuable in industrial applications, where workflows often begin with a predefined basic 3D model provided as a structural starting point. Artists or designers then refine this model by adding intricate details, textures, and stylistic elements based on reference images. Maintaining the basic 3D structure while enhancing visual details is often more practical than achieving perfect alignment with reference images. IPDreamer simplifies this process by enabling efficient editing of coarse 3D objects, reducing the need for extensive manual refinement and allowing for faster deployment in industrial pipelines.
>
> We appreciate your feedback and have included a discussion about potential improvements in alignment accuracy in our Future Work section (Appendix A.4).
>
> **Q3: The masks in Fig.4 are not quite aligned with the corresponding parts.**
>
> **A3:** Thanks for your valuable question. In the original paper, the masks shown in Fig. 4 were chosen to clearly demonstrate which parts the textual prompts focus on, instead of using strict binary (0-1) masks. In the actual optimization process, we utilize strict binary (0-1) masks for the Mask-guided Compositional Alignment strategy. We have updated Fig. 4 in the revised paper to reflect these binary masks for better clarity and accuracy.
>
> **Q4: It's hard to see the effectiveness of mask-guided compositional alignment.**
>
> **A4:** Thank you for raising this important concern. The effectiveness of mask-guided compositional alignment is most evident in challenging scenarios, specifically:
> 1. When there are significant disparities in both appearance and semantics between the reference image prompts and the initialized 3D objects
> 2. When multiple complex images are used as input conditions for 3D synthesis
>
> As demonstrated in Fig. 3, Fig. 6(a), and Fig. 11, without mask-guided compositional alignment, the generated 3D objects are unreasonable. In contrast, with our proposed alignment approach, the results show marked improvement in quality and coherence. These comparisons directly illustrate the crucial role of mask-guided compositional alignment in achieving high-quality 3D synthesis.
> Please feel free to ask if you need any further clarification.

---

> > ### Author Response · Authors · 2024-11-19
> > **Response to Reviewer PRnJ (Part 2/2)**
> >
> > **Q5: The results provided in Fig. 5 are not very good.**
> >
> > **A5:** Thank you for your suggestion. Fig. 5 showcases the results of IPDreamer achieving texture editing on 3D objects with complex images as references. The 3D objects generated by IPDreamer maintain the geometric structure and semantics of the initial 3D models while aligning well with the features of the reference complex images, demonstrating that IPDreamer successfully utilizes complex images for 3D object texture editing. Regarding your concern, we would appreciate it if you could specify in which aspects Fig. 5 falls short—whether it is in terms of generation quality, alignment with the guiding images, or other factors. Knowing the specific issues will help us refine the results or include them in the limitations section of the paper, and we can work on improving this aspect in future work.
> >
> > **Q6: What if we apply the best text-to-2D diffusion model to the DreamFusion or other text-to-3D pipeline and carefully design the text prompts? For example, the text-to-2D diffusion model that's capable of generating complex and high-resolution images.**
> >
> > **A6:** Thank you for your very valuable question. In fact, the results of methods such as ProlificDreamer and Fantasia3D demonstrate that text-to-3D generation techniques are already capable of producing high-quality 3D objects. However, the main challenge with text-to-3D methods lies in the inability to control the appearance of the generated 3D output. Even though text-to-image generation models are powerful, and no matter how long or detailed the textual prompts may be, they still cannot provide the same level of detailed visual information as images. Therefore, the approach proposed in this paper utilizing image prompt adaptation to guide 3D synthesis is not only controllable but also highly flexible.

---

### Author Response · Authors · 2024-11-19
**Clarifications and Summary of Key Contributions**

We'd like to express our gratitude to the reviewers and ACs for their thorough evaluation and valuable feedback. To facilitate a clearer understanding of our work, we have prepared a summary of our key contributions and would like to address several important points that may have been misinterpreted during the review process.

**Contributions**:
1. For cases with unclear boundaries, such as "Leaves flying in the wind," IPDreamer can generate reasonable 3D objects based on these conditions.
2. IPDreamer can utilize complex images to achieve high-quality texture editing on coarse 3D objects.
3. The Mask-guided Compositional Alignment strategy introduced in IPDreamer enables the concurrent integration of multiple image prompts to effectively guide the optimization process of a single 3D object.

Our IPDreamer is capable of achieving the functions mentioned in the contributions, which are difficult for existing text-to-3D and single-image-to-3D methods to accomplish. Then, we would now like to address several points raised during the review process.

**Clarifications**:
1. In the original version of the paper, we compared our method with single-image-to-3D methods, such as LRM and LGM. In the revised paper, we add results from additional single-image-to-3D methods for further comparison.
2. Our method primarily transfers features from complex images to an initialized coarse 3D model for flexible and high-quality 3D object synthesis. It is important to note that in industry applications, it is common first to provide an initial 3D object and then perform texture editing on it. Therefore, IPDreamer has significant practical value for building 3D assets.
3. The Mask-guided Compositional Alignment scheme is essential when substantial semantic difference exists between the reference complex image prompt and the initialized 3D object, or when multiple complex images are used as input conditions. When the reference image and the initial 3D model do not have significant semantic differences, high-quality 3D object synthesis can be achieved using only equations (2) to (7), which demonstrates the robustness of IPSDS.

In response to the insightful feedback from our reviewers, we make minor adjustments to both the main content and the figures and add more analysis in the Appendix. All modified sections, including the updated explaination and figures, are highlighted in blue (with figure captions also marked in blue if any modifications were made).

---

> ### Author Response · Authors · 2024-11-25
> **Kindly Invitation to Join Discussions**
>
> Dear Reviewers,
>
> We extend our heartfelt gratitude to all reviewers and the Area Chair for their thorough evaluation and constructive feedback on our submission. In our response, we have diligently addressed all the reviewers' questions and incorporated their suggestions to adjust the main paper. Additionally, we have included some extra analyses in the appendix.
>
> We sincerely hope that our revisions address your concerns and meet your expectations. Your insights have greatly improved our work, and we welcome further suggestions to refine it. We look forward to engaging in meaningful discussions and are eager to address any remaining concerns.
>
> Thank you once again for your time and thoughtful comments.
>
> Best regards, \
> The Authors

---

### Author Response · Authors · 2024-11-30
**Clarifications and Contributions: A Letter to Area Chairs, Reviewers, and 3D Generation Researchers**

Dear Area Chairs, Reviewers, and 3D Generation Researchers,

We sincerely thank all the reviewers and area chairs for their valuable contributions to improving this paper. The purpose of this letter is twofold: on one hand, to assist the reviewers and area chairs in making their final judgments, and on the other hand, **since this work has genuinely helped us solve several practical problems, we hope this paper will assist researchers in the 3D generation field by addressing potential challenges and providing some ideas.** Here, we would like to highlight the motivation, contribution, application, and future work of our paper once again.

**Motivation**

Although we have already clearly stated the motivation in the introduction, we would like to restate it here to provide a more intuitive understanding of our starting point. This will focus on the limitations of text-to-3D and single-image-to-3D methods, and explain the core idea behind our IPDreamer.

- Text-to-3D has achieved impressive realism, but the generated results often lack precision and controllability due to the limited information provided by the input text.

- In contrast, single-image-to-3D methods benefit from the prior information contained in the input image, allowing for better control over the output appearance. However, as researchers familiar with this approach will know, these methods require the subject in the input image to be very clear and unobstructed, often needing to be segmented out from the background. This is a limitation since such images are difficult to obtain in practice, as most images contain occlusions, and the main subject may not be easily isolated. This makes single-image-to-3D methods more restrictive and less stable when combined with text-to-image methods for generating text-to-3D results.

- The core idea of our paper is to extract corresponding features from complex images and align them with 3D objects. This approach not only enables the generation of 3D results for ambiguous subjects, such as “leaves flying in the wind,” but also allows for high-quality 3D object texture editing.

**Contribution**

- First, with the proposed IPSDS and Mask-guided Compositional Alignment, our method IPDreamer can leverage complex images for texture editing of 3D objects, generating high-fidelity 3D results that preserve the initial shape of the 3D object while adapting its appearance based on the complex image. This functionality is further elaborated in the application section.

- Since we can guide the generated results with complex images, our method can handle testing samples that are ambiguous or lack a clear main subject. For instance, we can generate the ideal 3D results for vague descriptions like "leaves flying in the wind" or "splashing wave."

- By leveraging Mask-guided Compositional Alignment, IPDreamer can also use multiple complex images to collaboratively guide the optimization of a single 3D object.

**Application**

Next, we will discuss the practical value of our contributions, especially the texture editing.

- In real-world applications like 3D game character design or scene creation, a common workflow involves creating a base 3D structure, which is then edited by modelers based on designers' sketches. IPDreamer can significantly reduce the amount of work for the modelers.

- For chibi-style character design, where the 3D structure of the character is usually fixed, but the appearance (texture) needs modification, our high-quality texture editing using Mask-guided Compositional Alignment and IPSDS can perfectly achieve this.

- For fixed 3D results, our method can guide the generation of diverse 3D outputs by leveraging a variety of complex images. This is particularly useful for enhancing the diversity of 3D data.

- In academic research, for example, in 4D generation, where we need to generate significantly different appearances of the same person, our texture editing can complement deformation networks, which often struggle to create noticeable appearance changes.

There are many other potential applications, but here we have listed some of the most familiar ones for reference.

**Future Work**

The core idea behind IPDreamer is extracting features from complex image prompts and aligning them with 3D objects. Future work could focus on improving feature extraction and refining the alignment process to enhance the accuracy and versatility of this method.

Finally, we would like to express our sincere gratitude once again to the reviewers and area chairs for their efforts. We deeply appreciate the constructive suggestions that have helped us improve the paper. **We respect the reviewers' opinions and fully support the area chairs' final decision. At the same time, we hope that the problems solved by IPDreamer will help many others and inspire new ideas in the 3D generation community.**

Sincerely, \
The Authors

---

> ### Author Response · Authors · 2024-12-02
> **Kindly Invitation (Last Day)**
>
> Dear Reviewers,
>
> We would like to once again express our sincere gratitude to the AC and all the reviewers for their valuable suggestions, which have helped us significantly improve the paper. We are confident that our work will positively contribute to the 3D community. Therefore, after the final decision, we plan to share the OpenReview link of this paper with as many relevant researchers as possible. Our Q&A record will help those unfamiliar with this paper better understand its contributions and potentially inspire new ideas.
>
> Therefore, we invite the reviewers to feel free to ask any further questions about details that might still be unclear, and we will be happy to provide answers before the end of the discussion.
>
> Once again, we sincerely thank the AC and all the reviewers for their patience and support. We respect and support everyone's judgment.
>
> Sincerely, \
> The Authors

---

> > ### Author Response · Authors · 2024-12-03
> > **Kindly Final Response**
> >
> > Dear Area Chair and Reviewers,
> >
> > We would like to express our sincere gratitude for the patience and assistance provided by all the Area Chairs and Reviewers. As we did not receive any additional questions on the final day, we would like to take this opportunity, before the meta-review begins, to once again briefly address the last two concerns raised by the reviewers to aid in the final judgment:
> >
> > - **Regarding Texture Editing**: In the main paper, we have thoroughly demonstrated the input conditions for this task and the high-quality generation results. Additionally, in the application section of the global letter, we have explained the rationale and practical value of this task setup. Furthermore, existing methods struggle to achieve the same level of quality for this task. Therefore, our method's texture editing possesses significant application value and lacks suitable alternatives.
> >
> > - **Regarding Partial Image Segment**: We have illustrated in the paper that this is not a particularly challenging task. Due to the stability of the IPSDS and Mask-guided Compositional algorithms, we only need to obtain rough partial images to reliably achieve high-quality desired 3D synthesis. The core contributions of our work are the IPSDS and Mask-guided Compositional methods.
> >
> > Once again, we deeply appreciate the patience and support of the Area Chairs and Reviewers. We respect and support the final decision. However, **we have to respectfully point out that the remaining two concerns used as reasons for rejection seem somewhat strained.** The value of our work and the necessary supplementary information have been well presented in the revised paper.
> >
> > Sincerely, \
> > The Authors

---

### Meta-Review · Area_Chair_KaRv · 2024-12-24

**Metareview:**

This work proposes IPDreamer a novel framework for text-input or image-input based 3D generation using guidance from a diffusion model. The key contribution of this work is to allow for precise control of the appearance of the generated mesh via a provided high-quality prompt image or set of images. To enable this, the authors propose the Image Prompt Score Distillation Sampling (IPSDS) loss and a mask guided compositional alignment module which enables the alignment of a prompt image with significantly differing content from the mesh/input image or to multiple prompt images. The authors show various qualitative and quantitative comparisons to state of the art approaches and superior performance of their approach in comparison to them in precisely optimizing for both the generated object's fine-grained texture and geometry.

The strength of this paper is that it addresses a gap in the literature on 3D reconstruction in being able to edit the texture and fine grained shape of images using the style and appearance of high resolution prompt images. This is a practical workflow employed by digital artists in may 3D content creation tasks. Additionally the visual results of the proposed method are impressive.

On the flip side, there are legitimate concerns about the speed of the method, which is optimization-based in comparison to the more recent faster feedforward methods; the method's inability to segment the prompt images correctly into parts and the inability of the method to strictly follow the structure of the provided input image in all circumstances.

**Additional Comments On Reviewer Discussion:**

Four reviewers provided scores of 6, 3, 6, 5. The reviewers appreciated the novelty of the proposed task and that of its proposed solution, and the quality of the results. However, they expressed concerns about the speed of the method, which is optimization-based in comparison to the more recent faster feedforward methods; its inability to segment the prompt images correctly into parts sometimes; its inability to strictly follow the structure of the provided input image sometimes; the practical significance and relevance of the proposed task of prompting the 3D generation process with a set of high quality 2D texture images; and the lack of comparisons to some state-of-the-art methods. During the rebuttal phase, the authors diligently provided detailed responses to all of the reviewers' responses and also updated many sections of the paper and the supplement to improve its clarity and presentation. Several of the reviewers's concerns were successfully addressed, but the reviewers remained mixed.

Overall, the AC feels that the proposed method is relevant to the practical workflows that artists employ to create initial versions of 3D assets and to improve their detailed appearance and geometry. While the method is not perfect in preserving the shape of the original coarse 3D object and some prompt images may not be perfectly segment-able into parts, there are many other circumstances in which this method is indeed be valid and works well as shown by the numerous examples in the paper. Hence it has the potential to successfully speed up practical 3D content creation workflows in many circumstances. Hence, all things considered the AC leans towards accepting this work as it represents a significant novel contribution to the research community

---

### Decision · Program_Chairs · 2025-01-22

Accept (Poster)